# HOW WELL DOES PERSISTENT HOMOLOGY GENERALIZE ON GRAPHS?

## ABSTRACT

Persistent Homology (PH) is one of the pillars of topological data analysis that leverages multiscale topological descriptors to extract meaningful features from data. More recently, the combination of PH and neural networks has been successfully used to tackle predictive tasks on graphs. However, the generalization capabilities of PH on graphs remain largely unexplored. To bridge this gap, we provide a lower bound on the VC dimension of PH on graphs. Further, we derive a PAC-Bayesian analysis for a popular PH-based classifier, namely PersLay, providing the first data-dependent generalization guarantee for PersLay. Notably, PersLay consists of a general framework that subsumes various vectorization methods of persistence diagrams in the literature. We substantiate our theoretical analysis with experimental studies and provide insights about the generalization of PH on real-world graph classification benchmarks.

## 1 INTRODUCTION

Topological Data Analysis (TDA) harnesses tools from algebraic topology to unveil the underlying shape and structure of data, and has recently gained significant traction within machine learning. One of its flagship methodologies is Persistent Homology (PH) (Edelsbrunner & Harer, 2008), which endeavors to characterize topological properties like connected components and loops within data. Remarkably, PH has found success across diverse scientific domains, spanning from computer vision (Hu et al., 2019) and drug design (Kovacev-Nikolic et al., 2016) to fluid dynamics (Kramár et al., 2016) and material science (Lee et al., 2017).

PH has also emerged as a promising tool (Chen et al., 2021; Zhao et al., 2020; Rieck et al., 2019) to augment the representational capabilities of Graph Neural Networks (GNNs). GNNs are employed to encode graph-structured data, and have achieved state-of-the-art performance in various applications (Scarselli et al., 2009; Kipf & Welling, 2017; Hamilton et al., 2017; Veličković et al., 2018; Xia et al., 2021). Many popular GNNs implement local message-passing schemes that are known to have limited expressivity (Garg et al., 2020); in particular, these GNNs are bounded in power by the Weisfeiler-Leman (WL) test for isomorphism (Xu et al., 2019; Morris et al., 2019; Maron et al., 2019; Joshi et al., 2023). PH has been employed to furnish global structural signatures that go beyond what can be captured by message-passing schemes, so PH has been integrated into GNNs to enhance their expressivity (Carrière et al., 2020; Hofer et al., 2017; Zhao & Wang, 2019).

Expressivity of machine learning models is certainly important (Sato, 2020); however, it does not guarantee that powerful models that do well on training data would *generalize*, i.e., predict well on unseen data as well. In fact, expressivity and generalization can be at odds with each other, and finding a good tradeoff holds the key to success of machine learning models (Garg et al., 2020). The importance of generalization bounds cannot be overstated, as they play a pivotal role in ensuring the reliability and applicability of machine learning models (Nia et al., 2023). In this context, there are two fundamental approaches to proving these bounds: data-independent and data-dependent (Sefidgaran & Zaidi, 2023), each offering unique insights into the generalization problem.

Recently, several works have analyzed the generalization ability of GNNs in the context of various prediction tasks for graphs (Scarselli et al., 2018; Verma & Zhang, 2019; Garg et al., 2020; Liao et al., 2020; Maskey et al., 2022; Esser et al., 2021; Zhou et al., 2022; Ju et al., 2023). In contrast, theoretical underpinnings underlying the success of TDA methods are underexplored; in particular, generalization capabilities of PH methods remain largely uncharted territory.

## 1.1 Our contributions

We fill the gap concerning the generalization behavior of PH from both data-dependent and data-independent perspectives. Specifically, we first lower-bound the VC dimension of PH for graph-structured data. VC dimension is a classic measure for model complexity, often used to guide model selection in machine learning settings, and is intimately related to the framework of PAC (probably approximately correct) learning theory (Vapnik & Chervonenkis, 1971).

We then establish the first data-dependent PAC-Bayesian generalization bounds (McAllester, 1999; 2003; Dziugaite & Roy, 2017) for PH, focusing on a versatile method *PersLay* that leverages *extended persistence* to effectively represent detailed topological features (Carrière et al., 2020). Our PAC-Bayes bound considers persistence diagrams obtained from fixed filtration functions, and therefore applies more generally to topological objects other than graphs. Our analysis hinges on some key technical insights and arguments, beyond a typical PAC-Bayes analysis, to overcome challenges arising from the inherent heterogeneity of the PersLay model.

We provide strong empirical evidence on five standard real-world datasets to substantiate the merits of our analysis. First, our experiments confirm strong correlation between observed generalization performance and the expected behavior as a function of different parameters in our bounds. Then, we use our bounds to propose a regularized version of PersLay with demonstrable empirical benefits.

## 1.2 Related works

**Expressivity and generalization of GNNs.** Xu et al. (2019); Maron et al. (2019) analyzed the representational power of GNNs in terms of the 1-WL test, revealing theoretical limits on their expressivity (Garg et al., 2020). This has motivated a surge of works aiming to go beyond 1-WL with GNNs (e.g., Li et al., 2020). Regarding generalization, Scarselli et al. (2018) first provided upper bounds on the order of growth of VC-dimension for GNNs. Garg et al. (2020) presented data-dependent generalization bounds via Rademacher complexity. Recently, Morris et al. (2023) employed the WL test alongside VC-dimension to gain insights about the generalization performance of GNNs. For details about the expressivity and learning of GNNs, we refer to Jegelka (2022).

**Expressivity of PH.** Rieck (2023) discussed the expressivity of PH on graphs, showing that PH is at least as powerful as the $k$-FWL (Folklore WL) test (Cai et al., 1992). Immonen et al. (2023) characterized the class of graphs that can be recognized by methods that rely on color-based filtrations, and proposed a more expressive method that combines vertex- and edge-level filtrations. However, to the best of our knowledge, there are no works concerning the generalization of PH-based methods.

**Learning theory and PH.** Birdal et al. (2021); Dupuis et al. (2023) explored a connection between learning theory and TDA and analyzed generalization error in terms of the so-called 'persistent homology dimension'. Chen et al. (2018) used the topological complexity of a model as a regularization term in order to simplify the model's topological complexity without sacrificing its flexibility.

**PAC-Bayes.** The PAC-Bayes framework (McAllester, 1999; 2003) allows us to leverage knowledge about learning algorithms and distributions over the hypothesis set for achieving tighter generalization bounds. Remarkably, Neyshabur et al. (2018) presented a generalization bound for feedforward networks in terms of the product of the spectral norm of weights using a PAC-Bayes analysis. For GNNs, Liao et al. (2020) exploited PAC-Bayes to improve over the generalization bound by Garg et al. (2020). Dziugaite & Roy (2017) optimized the PAC-Bayes bound directly and obtained non-vacuous generalization bounds for deep stochastic neural network classifiers.

## 2 Preliminaries

This section overviews PH on graphs (Edelsbrunner & Harer, 2008; Cohen-Steiner et al., 2009; Rieck, 2023), PersLay (Carrière et al., 2020), hinge loss, and PAC-Bayes (McAllester, 2003). Also, we provide in Table 3 (Appendix A) a summary of the notation adopted throughout the paper.

**Persistence Theory & Graphs**. We consider arbitrary graphs denoted as $G = (V, E)$, with vertex set $V = \{1, 2, ..., n\}$, and edge set $E \subseteq V \times V$. The set of all graphs we consider is denoted as $\mathcal{G}$.

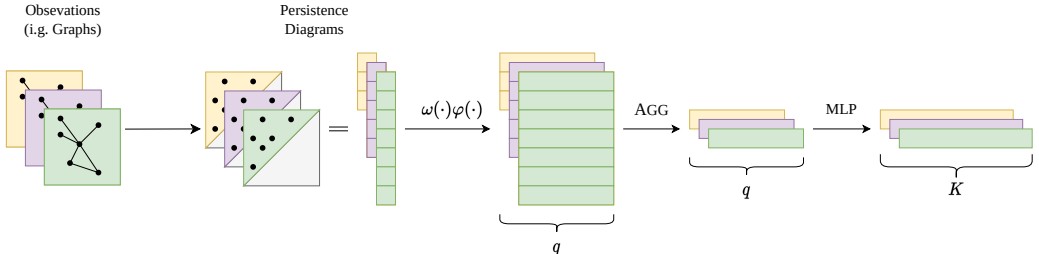

Figure 1: **PersLay Classifier**. Each graph is encoded as a persistence diagram, which is then embedded in some vector space using two functions $\omega, \varphi$ that are optimized during training. The resulting vectors are combined using a fixed permutation-invariant operator (i.e., AGG) and fed into an MLP.

Since graphs can be represented as topological spaces, Persistence Homology analysis can be applied to them. We call a finite nested sequence of subgraphs of $G$, i.e., $G_1 \subseteq G_2 \subseteq ... \subseteq G$ as a *filtrtaion* of a graph $G$. While various types of filtrations can be constructed (Hofer et al., 2017), a common choice involves utilizing a vertex *filtration function* $f : V \mapsto \mathbb{R}$ that allows us to obtain a permutation $\pi$ of the $n$ vertices such that $f(\pi(1)) \leq f(\pi(2)) \leq ... \leq f(\pi(n))$. Subsequently, a filtration induced by $f$ is an indexed collection $\{G_{f(\pi(i))}\}_{i=1}^n$, where each $G_{f(\pi(i))} \subseteq G$ is the subgraph with vertices $V = \{v \in V \mid f(v) \leq f(\pi(i))\}$. In the context of graphs, Persistence Homology typically tracks either the number of connected components or independent cycles (which correspond to 0- and 1-dimensional topological features) using efficient computational techniques. If a topological feature first appears in $G_{f(\pi(i))}$ and disappears in $G_{f(\pi(j))}$, then we encode its persistence as a pair $(f(\pi(i)), f(\pi(j)))$; if a feature does not disappear in $G_{f(\pi(n))} = G$, then its persistence is $(\cdot, \infty)$. The collection of all pairs forms a multiset that we call *persistence diagrams*. We denote the persistence diagram for a graph $G$ as $\mathcal{D}(G)$ and use card$(\mathcal{D}(G))$ to represent its *cardinality*, i.e., the number of (birth time, death time) pairs in $\mathcal{D}(G)$.

**PersLay.** Carrière et al. (2020) introduced a general way to vectorize *persistence diagrams*. In particular, given a persistence diagram $\mathcal{D}(G)$ for a graph $G$, PERSLAY computes

$$\text{PERSLAY}(\mathcal{D}(G)) = \text{AGG}\left(\{\!\{\omega(p)\varphi(p) \mid p \in \mathcal{D}(G)\}\!\}\right), \tag{1}$$

where AGG is any permutation invariant operation (e.g., minimum, maximum, sum, or kth largest value), $\omega : \mathbb{R}^2 \mapsto \mathbb{R}$ is a *weight* function for the persistence diagram points, and $\varphi : \mathbb{R}^2 \mapsto \mathbb{R}^q$ is the so-called *point transformation*. More specifically, given a persistence pair $p \in \mathbb{R}^2$, PersLay introduces the *triangle point transformation* ($\Lambda$):

$$\varphi_\Lambda(p) = [\Lambda_p(t_1), ..., \Lambda_p(t_q)]^T, \quad \text{where} \quad \Lambda_p(t) = \max\{0, p[2] - |t - p[1]|\}, \quad t \in \mathbb{R}; \tag{2}$$

the *Gaussian point transformation* ($\Gamma$):

$$\varphi_\Gamma(p) = [\Gamma_p(t_1), ..., \Gamma_p(t_q)]^T, \quad \text{where} \quad \Gamma_p(t) = \exp\left(-\frac{|t - p|_2^2}{2\tau^2}\right), \quad t \in \mathbb{R}^2; \tag{3}$$

and the *line point transformation* ($\Psi$):

$$\varphi_\Psi(p) = [\Psi_p(t_1), ..., \Psi_p(t_q)]^T, \quad \text{where} \quad \Psi_p(t) = t[1]p[1] + t[2]p[2] + t[3], \quad t \in \mathbb{R}^3, \tag{4}$$

where $t_1, \ldots, t_q$ are learnable parameters.

The architectural design of PERSLAY is quite versatile and accommodate a wide range of traditional persistence diagram vectorizations, including persistence landscapes (Bubenik, 2015), persistence silhouette (Chazal et al., 2014), persistence images (Adams et al., 2016), and other Gaussian-based kernel approaches (Kusano et al., 2016; Le & Yamada, 2018; Jan Reininghaus et al., 2015).

**PersLay Classifier (PC).** For classification tasks, Carrière et al. (2020) combine PersLay with a feedforward network — multilayer perceptrons (MLPs) with 1-Lipschitz activations. We denote the overall model as PC $: \mathcal{G} \mapsto \mathbb{R}^K$, where $K$ denotes the number of classes. In particular, the PersLay classifier first computes a persistence diagram using an arbitrary fixed vertex-based

filtration function and then applies PERSLAY. After that, it employs $l$ times a nonlinear activation function followed by a linear layer. Figure 1 shows the architecture of the PersLay Classifier. We denote the activation function before layer $i$ by $\psi_i$ and the weights of the linear layers as $W_1 \in \mathbb{R}^{h_1 \times h_2}, ..., W_l \in \mathbb{R}^{h_l \times h_{l+1}}$ where $h_1 = q$ (PERSLAY output's width) and $h_{l+1} = K$.

Henceforth, we use $W^\varphi = vec\{t_1, ..., t_q\}$ and $W^\omega$ to represent the parameters (collected as vectors) of the point transformations and weight functions, respectively. Similarly, the parameters of the PC model are denoted by $w = vec\{W_1, ..., W_l, W^\varphi, W^\omega\}$.

**Margin-based loss.** Following Neyshabur et al. (2018); Liao et al. (2020), we consider the multi-class $\gamma$-margin loss. Let $(G, y) \in \mathcal{G} \times \{1, 2, \ldots, K\}$ denote a labeled graph (i.e., a pair graph-label), and $S$ denote a collection of $m$ labeled graphs sampled i.i.d. from some unknown distribution $D$. Then, the empirical error of a hypothesis $g_w : \mathcal{G} \to \{1, 2, \ldots, K\}$ with parameters $w$ is defined as

$$L_{S,\gamma}(g_w) = \frac{1}{m} \sum_{(G,y) \in S} \mathbf{1}\left(g_w(G)[y] \leq \gamma + \max_{j \neq y} g_w(G)[j]\right), \tag{5}$$

where $\gamma \geq 0$. Accordingly, we can define the generalization error as

$$L_{D,\gamma}(g_w) = \mathbb{P}_{(G,y) \sim D}\left(g_w(G)[y] \leq \gamma + \max_{j \neq y} g_w(G)[j]\right). \tag{6}$$

Note $L_{S,0}$ and $L_{D,0}$ are the empirical and the expected classification errors (0-1 loss), respectively.

**PAC-Bayesian analysis** adopts a Bayesian approach to the PAC learning framework (Valiant, 1984; McAllester, 1999; 2003; Langford & Shawe-Taylor, 2002). The idea consists of placing a prior distribution $P$ over our hypothesis class and then use the training data to obtain a posterior $Q$, i.e., the learning process induces a posterior distribution over the hypothesis class. In this setting, we define the empirical and generalization errors of $Q$ as $L_{S,\gamma}(Q) = \mathbb{E}_{w \sim Q}[L_{S,\gamma}(g_w)]$ and $L_{D,\gamma}(Q) = \mathbb{E}_{w \sim Q}[L_{D,\gamma}(g_w)]$, respectively. Importantly, we can leverage the Kullback-Leibler (KL) divergence between $Q$ and $P$ to bound the difference between the generalization and empirical errors (McAllester, 2003).

To compute PAC-Bayes bounds for models like neural networks, we can i) choose a prior, ii) apply a learning algorithm; and iii) add random perturbations (from some known distribution) to the learned parameters such that we ensure tractability of the KL divergence. Following this recipe, Neyshabur et al. (2018) introduced the important result in Lemma 1.

**Lemma 1** (Neyshabur et al. (2018)). *Let $g_w(x) : \mathcal{X} \to \mathbb{R}^k$ be any model with parameters $w$, and let $P$ be any distribution on the parameters that are independent of the training data. For any $w$, we construct a posterior $Q(w + u)$ by adding any random perturbation $u$ to $w$, s.t., $P(\max_{x \in \mathcal{X}} |g_{w+u}(x) - g_w(x)|_\infty < \frac{\gamma}{4}) > \frac{1}{2}$. Then, for any $\gamma, \delta > 0$, with probability at least $1 - \delta$ over an i.i.d. size-$m$ training set $S$ according to $D$, for any $w$, we have:*

$$L_{D,0}(g_w) \leq L_{S,\gamma}(g_w) + 4\sqrt{\frac{D_{KL}(Q(w+u)||P) + \log \frac{6m}{\delta}}{m-1}}$$

Lemma 1 tells us that if we have prior and posterior distributions and guarantees that the change of the model's output due to perturbations over the learned parameters is small with high probability, we can obtain a generalization bound. Leveraging this, Neyshabur et al. (2018) and Liao et al. (2020) worked out generalization upper bounds for feedforward networks and GNNs, respectively. This Lemma is also *key* for our further analysis.

## 3 GENERALIZATION BOUNDS

In this section, we build upon results on the expressivity of GNNs and PH (i.e., a model that separates two graphs based on their persistence diagrams obtained from arbitrary filtration functions) to introduce a lower bound on the VC dimension of PH. Albeit important, this bound does not take into account the underlying data distribution. Thus, we also develop a data-dependent PAC-Bayes upper bound on the generalization of the PersLay Classifier. Finally, we discuss our PAC-Bayes bound in light of other bounds obtained for GNNs and MLPs, and show how to leverage it as a regularizer. The overall dependence structure of our theoretical results is summarized in Figure 2.

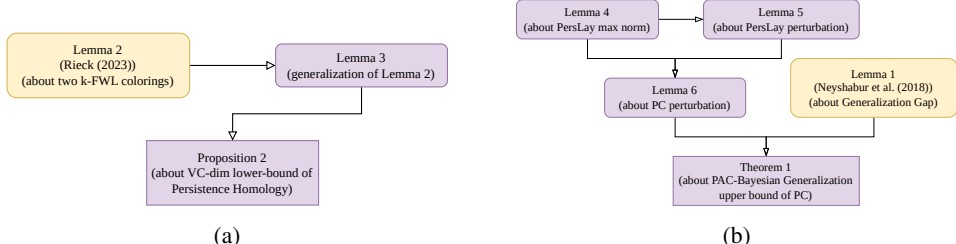

(a)                                                                           (b)

Figure 2: a) Connection between lemmas for VC-dimension analysis. b) Connection between lemmas for PAC-Bayesian analysis. Our contribution is colored in purple and previous work in yellow.

## 3.1 VC-DIMENSION

We now consider a *lower bound* result regarding the generalization ability of persistence homology. To do so, we provide an analogous proposition to a recent result by Morris et al. (2023) which connects the Weisfeiler-Leman algorithm (Weisfeiler & Leman, 1968) to the VC-dimension of message-passing neural networks (MPNNs, Xu et al., 2019), or simply GNNs.

**Proposition 1** (Proposition 2 in (Morris et al., 2023))**.** *Let $m_{n,d,L}$ be the maximal number of graphs with $d$-dimensional boolean features and at most $n$ vertices distinguishable after $L$ iterations of 1-WL. Then,* $\text{VC-DIM}(\text{GNN}(L)) \geq m_{n,d,L}$*, where $\text{GNN}(L)$ is an $L$-layer GNN model.*

We provide an analogous proposition for persistence homology, denoted by a model called `PH` that distinguishes graphs by comparing their persistence diagrams obtained from arbitrary filtration functions. Importantly, our result in Proposition 2 illustrates the inherent tension between generalization and expressivity of any PH method on graphs (both can be lower bounded in terms of the WL hierarchy). In particular, enhancing expressivity leads to an increase in the VC-dimension, thereby worsening the ability to generalize.

**Proposition 2.** *Let $m'$ be the maximal number of distinguishable graphs (with at most $n$ nodes) by $k$-FWL, then $\text{VC-DIM}(PH) \geq m'$.*

To prove this statement, we first generalize Lemma 5 by Rieck (2023) (details in the Appendix).

**Lemma 2** (Lemma 5 in Rieck (2023))**.** *Given $k$-FWL colorings of two graphs $G$ and $G'$ that are different, there exists a filtration of $G$ and $G'$ such that the corresponding persistence diagrams in dimension $k - 1$ or dimension $k$ are different.*

**Lemma 3** (Generalization of Lemma 5 in (Rieck, 2023))**.** *Given $k$-FWL colorings of graphs $G_1, ..., G_n$ that are distinct, there exists a filtration such that persistence diagrams computed using this filtration function of $G_1, ...G_n$ are distinct in dimension $k - 1$ or dimension $k$.*

Now, using this generalization, we are able to prove Proposition 2.

*Proof.* If we can distinguish $m'$ graphs using $k$-FWL, it means that the obtained colorings are distinct for these $m'$ graphs. Hence, by Lemma 3, there exists filtration such that persistence diagrams are distinct in the dimension $k - 1$ or $k$ for these $m'$ graphs; so, we can shatter them using PH. □

## 3.2 PAC-BAYES BOUNDS FOR CLASSIFICATION WITH PERSLAY

We now introduce a PAC-Bayes upper bound for the PersLay Classifier (PC). Proofs can be found in the Appendix. Here, we make the following assumptions (also common in the literature):

1. Data, i.e., tuples $(G, y)$, are i.i.d samples drawn from some unknown distribution $D$;
2. The maximum hidden dimension across all layers is $h$;
3. $\forall 1 \leq i \leq l \quad \text{Lip}(\psi_i) \leq 1, \psi_i(x) \leq |x|$. Thus our analysis subsumes any combination of most commonly used non-linear activation functions (e.g., ReLU, Leaky ReLU, SoftPlus, Tanh, and Sigmoid) in the MLP;
4. The norm of the elements of *persistence diagram* are contained in a $\ell_2$-ball with a radius $b$;

5. All of the considered point transformations and weight functions are Lipschitz continuous with respect to the parameters.

6. $\forall \omega : \mathbb{R}^2 \mapsto \mathbb{R} \ \forall p \in \mathcal{D} \ |\omega(p)| < 1$. We can achieve this by output normalization;

7. We consider only parameter-independent filtration functions. They might use vertex features or hyperparameters but no learnable parameters (e.g., as in (Hofer et al., 2020)).

We begin by introducing results concerning perturbation bounds of PERSLAY. In particular, Lemma 4 estimates the maximum output norm of PERSLAY layers.

**Lemma 4.** *Let $W^\varphi$ be the parameters of the point transformation in* PERSLAY. *Then, we have that*

$$\forall G \in \mathcal{G} \quad |\text{PERSLAY}_w(\mathcal{D}(G))|_2 \leq B_1 |W^\varphi|_2 + C_1 \leq M_1(|W^\varphi|_2 + 1)$$

*where*

$$A_1 = \begin{cases} \max_{G \in \mathcal{G}} card(\mathcal{D}(G)) & \text{if } \text{AGG} = \text{sum} \\ 1 & \text{if } \text{AGG} = \text{mean } \text{or } k\text{-max} \end{cases}, \quad (B_1, C_1) = \begin{cases} (\sqrt{2}, \sqrt{8q}\, b) & \text{if } \varphi = \Lambda \\ (0, \sqrt{q}) & \text{if } \varphi = \Gamma \\ (\sqrt{3}(b+1), 0) & \text{if } \varphi = \Psi \end{cases}$$

*and $M_1 = A_1 \max\{B_1, C_1\}$.*

The next result (Lemma 5) helps us to upper bound the difference in the outputs of PERSLAY under a slight perturbation of its weights. We note that the analyses in Lemmas 4 and 5 preserve the inherent flexibility of PERSLAY and ensure the generality of our results.

**Lemma 5.** *Let $w = vec\{W^\varphi, W^\omega\}$ and $u = vec\{U^\varphi, U^\omega\}$, where $U^\varphi$ and $U^\omega$ denote small perturbations on the parameters of the point transformations and weight functions of* PERSLAY, *respectively. Then:*

$$|\text{PERSLAY}_w(\mathcal{D}(G)) - \text{PERSLAY}_{w+u}(\mathcal{D}(G))|_2 \leq M_2(|W^\varphi|_2 + 1)(|U^\varphi|_2 + |U^\omega|_2)$$

*where*

$$A_2 = \begin{cases} \max_{G \in \mathcal{G}} card(\mathcal{D}(G)) & \text{if } \text{AGG} = \text{sum} \\ 3 & \text{if } \text{AGG} = \text{mean } \text{or } k\text{-max} \end{cases}, \quad B_2 = \begin{cases} 1 & \text{if } \varphi = \Lambda \\ \frac{1}{\tau e^{1/2}} & \text{if } \varphi = \Gamma \\ \sqrt{3}(b+1) & \text{if } \varphi = \Psi \end{cases},$$

$M_2 = A_2 \max\{B_2, M_1 Lip(\omega)\}$, and $Lip(\omega)$ is a Lipschitz constant of $\omega$.

Finally, the following Lemma encompasses the previous Lemmas and composes them with linear layers bounds introduced by Neyshabur et al. (2018).

**Lemma 6.** *Let $w = vec\{W_1, ..., W_l, W^\varphi, W^\omega\}$ and $u = vec\{U_1, ..., U_l, U^\varphi, U^\omega\}$, where $U_i$ is the perturbation of $i$th linear layer of* PC, *$U^\varphi$ is the perturbation of the point transformation part, and $U^\omega$ is the perturbation of the weight part of* PERSLAY. *Also, let $T \geq \max\{||W_1||_2, ..., ||W_l||_2, |W^\varphi|_2 + 1\}$ and $\forall i : ||U_i||_2 \leq \frac{1}{l}T$, then we can derive the following upper bound:*

$$|\text{PC}_w(x) - \text{PC}_{w+u}(x)|_2 \leq eM\, T^{l+1}\left(|U^\varphi|_2 + |U^\omega|_2 + \sum_{i=1}^{l} ||U_i||_2\right)$$

*where $M = \max\{M_1, M_2\}$ from Lemmas 4 and 5 and $e$ is the Euler's constant.*

*Proof sketch.* We prove this lemma using induction by the number of linear layers as in (Neyshabur et al., 2018). The main difference is the $l = 0$ case: we use Lemma 4 and Lemma 5 to prove the statement. For the transition $l \to l + 1$ we simply use the definition of the linear layer: $\text{Linear}(x) = W(\psi(x))$. Another difference between this proof and one from (Neyshabur et al., 2018) is that weights of the PERSLAY during the derivation appear in the brackets with perturbation weights. So, we need the variable $T$ to upper bound all the weights simultaneously to get a nice-looking overall expression. Otherwise, we will not be able to use Lemma 1 as smoothly as they used it. $\square$

Theorem 1 provides the bound on the generalization gap of the PersLay Classifer.

Table 1: The dependence of the PAC-Bayesian bounds on width and spectral norm of weights for different models. Here, $h$ is the maximum width across layers; $W_i$ are the weights of $i$-th linear layer; and $l$ is the depth (number of layers). The 1st row follows from our bound (Theorem 1), with $w$ denoting the parameters of the PersLay classifier and $q$ the dimension of PersLay embeddings. The 2nd row shows the result by Neyshabur et al. (2018), where $||\cdot||_F$ denotes the *Frobenius norm*. The 3rd row shows the result by Liao et al. (2020), where $d$ is the maximum degree across graphs.

| Classifier (Reference) | Width, $h$ | Weights spectral norm, $W_i$ |
|---|---|---|
| PERSLAY Classifier (this work) | $\mathcal{O}\left(\sqrt{qh\ln h} + \ln q\right)$ | $\mathcal{O}\left(\sqrt{l^2 \ln (l)|w|_2^2 \beta^{2(l+1)}}\right)$ |
| Feedforward networks (Neyshabur et al., 2018) | $\mathcal{O}(\sqrt{h\ln h})$ | $\mathcal{O}\left(\sqrt{l^2 \ln (l) \prod_{i=1}^{l} ||W_i||_2^2 \sum_{i=1}^{l} \frac{||W_i||_F^2}{||W_i||_2^2}}\right)$ |
| Message-passing GNNs (Liao et al., 2020) | $\mathcal{O}(\sqrt{h\ln h})$ | $\mathcal{O}\left(\sqrt{d^{l-1}l^2 \ln (l) \prod_{i=1}^{l} ||W_i||_2^2 \sum_{i=1}^{l} \frac{||W_i||_F^2}{||W_i||_2^2}}\right)$ |

**Theorem 1.** *Let $w = vec\{W_1, ..., W_l, W^\varphi, W^\omega\}$ and $M = \max\{M_1, M_2\}$ from Lemmas 4 and 5. Then for any $\gamma, \delta > 0$ with probability at least $1 - \delta$ over i.i.d size-m training set S according to D, for any $W_1, ..., W_l, W^\varphi, W^\omega$, we have:*

$$L_{D,0}(\text{PC}_w) \leq L_{S,\gamma}(\text{PC}_w) + \mathcal{O}\left(\sqrt{\frac{l^2 M^2 h \ln (lh)|w|_2^2 \beta^{2(l+1)} + \ln \frac{lMm}{\delta}}{\gamma^2 m}}\right)$$

*where $\beta = \max\{||W_1||_2, ..., ||W_l||_2, |W^\varphi|_2 + 1\}$*

*Proof sketch.* At first, we construct a function of weights, $\beta = \max\{||W_1||_2, ..., ||W_l||_2, |W^\varphi|_2 + 1\}$. Then we fix arbitrary $\hat{\beta}$ and consider $w$ such that $|\beta - \hat{\beta}| \leq \frac{1}{l}\beta$ and choose $\sigma$ which depends on $\hat{\beta}$, so that we can apply Lemma 6 and 1. Finally, we apply the union-bound argument to cover all possible $\beta$ and $w$, respectively. $\square$

### 3.3 DISCUSSION

**Heterogeneity of PC.** It is essential to underscore the fact that the described PC model is not a homogeneous entity. This model is a composition of PERSLAY layer and linear layers, and this very heterogeneity inherently prevents us from the application of elegant constructions that involve normalized weights (Neyshabur et al., 2018). This feature, in turn, introduces many challenges when analyzing its generalization behavior using the PAC-Bayesian framework. Namely, there are several conditions to satisfy when choosing $\beta$: it should be such that we can upper bound all appearances of model weights in the PC's perturbation and be able to apply Lemma 6, i.e., be able to satisfy $||U_i||_2 \leq \frac{1}{l}T$. It turns out to be non-trivial to satisfy both these conditions in the general case.

**Generality of PC.** Another important facet of this research is its all-encompassing analysis, delving into many scenarios that encompass the selection of *weight* and *point transformation* functions. This allows for a more profound comprehension of the PC method's performance across diverse settings. We believe it is important, especially considering that the freedom to choose the *weight* and *point transformation* functions was a pivotal decision-making point in the work presented in (Carrière et al., 2020). These challenges and their subsequent exploration lay at the very heart of our research, shedding light on the intricate interplay between the various components of the PC model.

**Influence of PersLay components.** Our analysis shows that when AGG = sum, it is hard to obtain reasonable generalization guarantees since $M$ depends on the cardinality of the persistence diagram, which can be large. Also, we can conclude that the generalization performance of PC using the line point transformation has a weaker dependence on $q$ ($M_1, M_2$ does not depend on $q$ in this case) than other point transformations. This is particularly relevant if one chooses a large value for $q$.

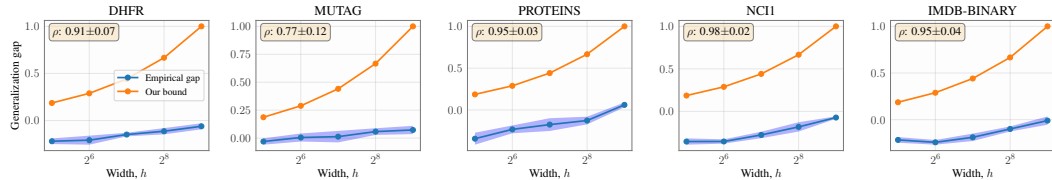

Figure 3: **Spectral norm vs. generalization gap**. Overall, our bound on the spectral norm of the weights is highly correlated with the generalization gap.

Figure 4: **Width vs. generalization gap.** The dependence of the empirical gap on the model width is captured by our bound. We obtain high average correlation for all datasets.

**Comparison to other bounds.** By contrasting our result (in Theorem 1) with PAC-Bayes bounds for feedforward neural networks (Neyshabur et al., 2018) and GNNs (Liao et al., 2020), one notable observation is the resemblance of the dependence on $\gamma$, $m$ and $b$. Table 1 compares our bounds to previous ones with respect to their dependence on model weights and width. We note that the main sources of difference between our result and previous ones stem from the choice of the persistence diagram vectorization (Lemmas 4 and 5) and from the combination of perturbations in vectorization and linear layers (Lemma 6). Importantly, by analyzing the dependence on the width, we conclude that our upper bound increases from $\mathcal{O}(\sqrt{h \ln h})$ to $\mathcal{O}(h\sqrt{\ln h})$ when $q = \Theta(h)$. So, to produce a tighter generalization bound, we recommend choosing $q = o(h)$.

### 3.4 REGULARIZING PERSLAY

Notably, we can leverage our bound to introduce a spectral norm regularizer for the PersLay Classifier. In particular, we can train PC using a regularized loss

$$L_{S,\gamma} + \lambda \sqrt{l^2 \ln(l)|w|_2^2 \beta^{2(l+1)}},$$

where $\lambda$ is a hyper-parameter that balances the influence of the two terms. This is similar to a weight-decay regularization approach, with the spectral norm of weights appearing in $\beta$.

## 4 EXPERIMENTS

To demonstrate the practical relevance of our analysis, we now consider the generalization of PersLay on real-world datasets, and report results for regularized models based on our bounds. In particular, we conduct two main experiments. The first one aims to analyze how well our bounds capture generalization gaps as a function of model variables. The second assesses to which extent a structural risk minimization algorithm that uses our bound on the weights spectral norm improve generalization compared to empirical risk minimizers. We implemented all experiments using PyTorch (Paszke et al., 2017), and implementation details are given in the Appendix B.

**Datasets and evaluation setup.** We use five popular benchmarks for graph classification: DHFR, MUTAG, PROTEINS, NCI1, IMDB-BINARY, which are available as part of TUDatasets (Kersting et al., 2016). We use a 70/10/20% (train/val/test) split for all datasets when we perform model selection. In this case, after selecting the optimal hyper-parameters, we retrain the model using train+val data. Here, we consider PersLay models with constant weight functions and Gaussian point transformations. Also, we remove graph-level features originally employed by PersLay. Regarding filtration functions, we closely follow Carrière et al. (2020) and use Heat kernel signatures with parameter values equal to 0.1 or 10, depending on the dataset. We train the models for 3000 epochs using the Adam optimizer (Kingma & Ba, 2015). We run five independent runs with different seeds.

Figure 5: Illustration of the empirical vs theoretical gap for different point transformations.

**Dependence on model components**. Figure 3 shows the generalization gap (measured as $L_{D,0} - L_{S,\gamma=1}$) and the bound on the weights spectral norm (see Table 1) over the training epochs. To evaluate how well our bound captures the trend observed in the empirical gap, we compute Pearson correlation coefficients between the two sequences across different seeds and report their mean and standard deviation for each dataset. As we can see, the average coefficient is greater than $0.78$ for all benchmarks, indicating a good correlation.

Figure 4 shows the empirical gap and our estimated bound as a function of the model's width. Again, we compute correlation coefficients between the two curves and find they are highly correlated (with an average correlation above $0.91$ on $4$ out of $5$ datasets). These results validate that our theoretical bounds can capture the trend observed in the empirical generalization gap.

**Point transformations.** We also compare the theoretical bound with the empirical gap for different choices of point transformations. Figure 5 reports the results. We apply the same constant accross transformations such that the highest bound value equals one. Overall, the Triangle function produces higher bounds whereas that Line seems to be associated with smaller constant factors. For three datasets, the higher the observed gap, the higher the bound.

**Regularized PC.** We compare variants of the PersLay Classifier trained via ERM (empirical risk minimization) and its regularized version. Here, we consider models with $l = 1$ or $2$, selected via hold-out validation. Table 2 reports accuracy results (mean and standard deviations) computed over five runs. Overall, the regularized approach significantly outperforms the ERM variant despite the use of small-sized networks. On 4/5 datasets, PC with spectral norm regularization is the best model.

Table 2: Comparison of PersLay with and without spectral norm regularization. We report accuracy numbers (mean and standard deviation) computed over five independent runs.

| Method | DRFH | MUTAG | PROTEINS | IMDB-B | NCI1 |
|---|---|---|---|---|---|
| PersLay (ERM) | $0.71 \pm 0.04$ | $0.88 \pm 0.02$ | $0.65 \pm 0.03$ | $0.65 \pm 0.01$ | $0.68 \pm 0.01$ |
| PersLay (w/ Spectral Reg.) | $\mathbf{0.72} \pm 0.02$ | $\mathbf{0.94} \pm 0.01$ | $\mathbf{0.72} \pm 0.01$ | $\mathbf{0.70} \pm 0.01$ | $0.68 \pm 0.01$ |

## 5 CONCLUSION, BROADER IMPACT AND LIMITATIONS

We derive the first PAC-Bayesian generalization bound for neural networks based on persistent homology for graph learning. Notably, the framework analyzed (PersLay) offers a flexible and general way to extract vector representations from persistence diagrams. Due to this generality, our analysis covers several methods available in the literature. Our constructions involve a perturbation analysis of PersLay's parameters and linear layers, which imposes specific challenges as the resulting model is inherently heterogeneous. In addition, we discuss a VC-dim lower bound for PH in terms of the WL hierarchy for isomorphism tests on graphs. We also validate our analysis using real-world data, and show that the proposed bound can be used in the framework of structural risk minimization.

Despite the significance and novelty of our theoretical analyses, we would like to acknowledge a limitation in our study, specifically concerning the absence of parameterization for filtration functions. While we provide valuable insights and methodologies, we should underscore the need for future investigations to delve into more general analyses encompassing parametrized filtration functions.

By shedding new light on the generalization of machine learning models based on persistent homology, we hope to contribute to the community by providing key insights about the limits and power of these methods, paving the path to further theoretical developments on PH-based neural networks for graph representation learning.

## REPRODUCIBILITY STATEMENT

All experiments in this work are reproducible. Upon acceptance, we will make our code, configurations, seeds, and trained models available under the MIT License on GitHub.

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
