# A    NOTATION

Table 3 summarizes the main mathematical symbols and abbreviations used in this work.

Table 3: Summary of notation and abbreviations.

| Notation | Description |
|---|---|
| $G = (V, E)$ | arbitrary graph with vertices $V$ and edges $E$ |
| $\mathcal{G}$ | the set of all considered graphs |
| $\mathcal{D}(G)$ | persistence diagram of cardinality card$(\mathcal{D}(G))$ |
| $\omega(\cdot)$ | arbitrary *weight* function, $\mathbb{R}^2 \mapsto \mathbb{R}$ |
| $W^\omega$ | parameter *vector* of the $\omega$ (weight) function |
| $q$ | dimensionality of the PERSLAY embeddings |
| $\varphi(\cdot)$ | arbitrary *point transformation*, $\mathbb{R}^2 \mapsto \mathbb{R}^q$ |
| $\varphi_\Lambda, \varphi_\Gamma, \varphi_\Psi$ | Triangle/Gaussian/Line *point transformation* |
| $t_i$ | $i$th parameter value(s) of a point transformation — following Carrière et al. (2020) |
| $W^\varphi$ | vector comprising all parameters of the point transformation $\varphi$, i.e., $W^\varphi = vec(\{t_i\}_{i=1}^q)$ |
| AGG | arbitrary aggregation function |
| PERSLAY | a mapping from *persistence diagrams* to $\mathbb{R}^q$ (Equation 1) |
| $K$ | number of classes |
| MLP | multi-layer perceptron with non-linear 1-Lipschitz activation functions |
| $h_i, h$ | dimensionality of the input to the $i$-th MLP layer, and $h = \max\{h_1, ..., h_{l+1}\}$ |
| $\psi_i$ | non-linear 1-Lipschitz activation funcion before $i$th layer |
| $W_i$ | weight matrix of the $i$-th MLP layer, $\in \mathbb{R}^{h_i \times h_{i+1}}$ |
| PC | a mapping from $\mathcal{G}$ to $\mathbb{R}^K$ consisting of $\mathcal{D}(G) \to$ PERSLAY $\to$ MLP |
| $vec(\cdot)$ | function that converts its input to a single *vector* |
| $D$ | distribution over labeled graphs (i.e., graph-label pairs), data distribution |
| $S, m$ | $S$ is a training set consisting of $m$ labeled graphs, i.e., $S = \{(G_i, y_i)\}_{i=1}^m$ |
| $\gamma$ | margin scalar used in the margin-based loss |
| $L_{S,\gamma}(g_w)$ | empirical error ($\gamma$-margin loss) of a hypothesis $g$ (with parameters $w$) $S$ |
| $L_{D,\gamma}(g_w)$ | generalization error ($\gamma$-margin loss) of a hypothesis $g$ (with parameters $w$) on $D$ |
| $b$ | the maximum norm of the input to the feedforward network in the PersLay Classifier |
| $D_{\mathrm{KL}}$ | KL-divergence between two distributions |
| VC-DIM | Vapnik-Chervonenkis dimension |
| $k$-FWL | $k$-th order *Folklore* Weisfeller-Leman algorithm |
| $\|\cdot\|_2, |\cdot|$ | Euclidean norm of a *vector*, absolute value |
| $\|\cdot\|_2, \|\cdot\|_F$ | Operator / Frobenius norm of a *matrix* |
| $U_i, U^\varphi, U^\omega$ | perturbation of the parameters of the $i$th MLP layer / the $\varphi$ function / the $\omega$ function |
| $\beta, \hat{\beta}$ | $\max\{\|W_1\|_2, ...\|W_l\|_2, |W^\varphi|_2 + 1\}$ / arbitrary approximation of $\beta$ |

## B  SECTION 3 OMITTED MATERIALS

**Lemma 2** (Lemma 5 in Rieck (2023)). *Given $k$-FWL colorings of two graphs $G$ and $G'$ that are different, there exists a filtration of $G$ and $G'$ such that the corresponding persistence diagrams in dimension $k - 1$ or dimension $k$ are different.*

*Proof.* Here we provide the proof of this lemma from Rieck (2023) (Appendix D).

The main idea involves harnessing the colors of $k$-tuples. We first identify all colors $c_1, c_2, ...$ with natural numbers $1, 2, ...$. We then expand $G$ and $G'$ to a simplicial complex containing all $k$-tuples as faces. Moreover, we assign *all* simplices in dimensions less than or equal to $k - 2$ a weight of $0$. Each $(k - 1)$-simplex is assigned its color according to the respective $k$-FWL coloring. As a consequence of the *pairing lemma* of persistent homology, every $(k - 1)$-simplex is either a creator simplex or a destroyer simplex (Edelsbrunner & Harer, 2008).

We handle the case of *creator simplices* first. Each creator simplex gives rise to an essential persistence pair in the $(k - 1)$-dimensional persistence diagram. Each such pair is of the form $(i, \infty)$, where $c_i$ is a color according to $k$-FWL.

Each *destroyer simplex*, by contrast, destroys a topological feature created by a $(k - 2)$-simplex, i.e. a $(k - 1)$-tuples, resulting in a pair of the form $(\cdot, j)$, with again $c_j$ being the corresponding $k$-FWL color. This pair is part of a $(k - 2)$-dimensional persistence diagram. When dealing with tuples or simplices, there is always the risk of an off-by-one error. In this theorem, despite dealing with $k$-FWL, only the $(k - 1)$-simplices, which have $k$ vertices, will be relevant.

By assumption, the $k$-FWL colors of $G$ and $G'$ are different, so a color $c$ must exist whose count is *different* in $G$ and $G'$, respectively. Since the *sum* of all colors arising in the two types of persistence pairs above is the number of colors of $k$-tuples, there is either a difference in color counts in the $(k - 1)$-dimensional or the $(k - 2)$-dimensional persistence diagrams of $G$ and $G'$, showing that they are not equal. $\square$

**Lemma 3** (Generalization of Lemma 5 in (Rieck, 2023)). *Given $k$-FWL colorings of graphs $G_1, ..., G_n$ that are distinct, there exists a filtration such that persistence diagrams computed using this filtration function of $G_1, ... G_n$ are distinct in dimension $k - 1$ or dimension $k$.*

*Proof.* We simply repeat the proof of Lemma 2 word by word using $G_1, ..., G_n$ instead of $G$ and $G'$. $\square$

The following result helps us to upper bound the KL-divergence.

**Lemma 7.**
$$D_{\mathrm{KL}}(\mathcal{N}(w, \sigma^2 \boldsymbol{I}) \,||\, \mathcal{N}(0, \sigma^2 \boldsymbol{I})) = \frac{|w|_2^2}{2\sigma^2}$$

*Proof.* Let these distributions have the same dimension, $k$. Then, there is a common formula for a KL-divergence of two multivariate distributions.

$$D_{\mathrm{KL}}(\mathcal{N}(w, \sigma^2 \boldsymbol{I}) \,||\, \mathcal{N}(0, \sigma^2 \boldsymbol{I})) =$$
$$= \frac{1}{2}\left( \mathrm{tr}\left((\sigma^2 \boldsymbol{I})^{-1}(\sigma^2 \boldsymbol{I})\right) - k + (0 - w)^T (\sigma^2 \boldsymbol{I})^{-1}(0 - w) + \ln\left(\frac{\det \sigma^2 \boldsymbol{I}}{\det \sigma^2 \boldsymbol{I}}\right) \right) =$$
$$= \frac{|w|_2^2}{2\sigma^2}$$

$\square$

Our next lemma helps us to upper bound the PERSLAY's perturbation, when $\mathrm{AGG} = k\text{-max}$.

**Lemma 8.** *Let $X$ be an arbitrary finite set and $f, g : X \mapsto \mathbb{R}$. Then we can say that:*

$$\left| k\text{-}\max_{x \in X} f(x) - k\text{-}\max_{x \in X} g(x) \right| \leq 3 \max_{x \in X} |f(x) - g(x)|$$

Figure 6: Case $g(n(k)) > g(m(k)), f(n(k)) \geq f(m(k))$

Figure 7: Case $g(n(k)) > g(m(k)), f(n(k)) \leq f(m(k))$

*Proof.* Denote $n : \mathbb{N} \mapsto X$ by a function that maps natural number $k$ to an element of $X$ that would be on $k$th position in order sorted by $f$. Denote $m$ as an analogous function but for $g$. Then, we are interested in the following expression: $|f(n(k)) - g(m(k))|$. Let us rewrite it:

$$
\begin{aligned}
|f(n(k)) - g(m(k))| = |f(n(k)) - g(n(k)) + g(n(k)) - g(m(k))| \leq \\
\leq |f(n(k)) - g(n(k))| + |g(n(k)) - g(m(k))| \leq \\
\leq \max_{x \in X} |f(x) - g(x)| + |g(n(k)) - g(m(k))|
\end{aligned}
$$

Now, the task is to prove that $|g(n(k)) - g(m(k))| \leq 2 \max_{x \in X} |f(x) - g(x)|$

Let us consider four cases:

- $g(n(k)) > g(m(k))$ and $f(n(k)) \geq f(m(k))$ (Fig. 6). In this case $\exists i \in \mathbb{N}$ such that $f(n(i)) > f(n(k))$ and $g(n(i)) < g(m(k))$. Indeed, if none of the elements "to the right" of $n(k)$ moved "to the left" of $m(k)$, then "to the right" of $m(k)$, there are at least $n - k + 1$ elements; however, there are must be exactly $n - k$ elements.

$$
\begin{aligned}
|g(n(k)) - g(m(k))| = g(n(k)) - g(m(k)) \leq g(n(k)) - g(n(i)) \leq \\
\leq f(n(k)) + (\max_{x \in X} |f(x) - g(x)|) - g(n(i)) < \\
< f(n(i)) + (\max_{x \in X} |f(x) - g(x)|) - g(n(i)) < 2(\max_{x \in X} |f(x) - g(x)|)
\end{aligned}
$$

- $g(n(k)) > g(m(k))$ and $f(n(k)) \leq f(m(k))$ (Fig. 7)

$$
\begin{aligned}
|g(n(k)) - g(m(k))| = g(n(k)) - g(m(k)) \leq \\
\leq f(n(k)) + \left( \max_{x \in X} |f(x) - g(x)| \right) - g(m(k)) \leq \\
\leq f(m(k)) + \left( \max_{x \in X} |f(x) - g(x)| \right) - g(m(k)) \leq \\
\leq 2 \left( \max_{x \in X} |f(x) - g(x)| \right)
\end{aligned}
$$

- The rest of the cases can be handled analogously.

$\square$

The following lemma helps us to determine the distribution parameters for weight perturbation to satisfy the margin.

**Lemma 9.** *Let* $w = vec\{W_1, ..., W_l, W^\varphi, W^\omega\}$. *Let* $u = vec\{U_1, ..., U_l, U^\varphi, U^\omega\}$, *where* $U_i$ *is the perturbation of* $i$th *linear layer of* PC *and* $U^\varphi$ *is the perturbation of* point transformation *part and* $U^\omega$ *is the perturbation of* weight *part of* PERSLAY.

*Let* $\hat{\beta}, \gamma \in \mathbb{R}_+$ *and* $\beta = \max\{||W_1||_2, ..., ||W_l||_2, |W^\varphi|_2 + 1\}$.

*If* $w$ *such that* $|\beta - \hat{\beta}| \le \varepsilon\beta$ *and* $\beta \ge \left(\frac{\gamma}{2M}\right)^{\frac{1}{l+2}}$, $U_i \sim \mathcal{N}(0, \sigma^2 I), 1 \le i \le l$; $U^\omega, U^\varphi \sim \mathcal{N}(0, \sigma^2 I)$, *then* $\forall G \in \mathcal{G}$ *with probability* $\frac{1}{2}$ *the following inequality holds:* $|\text{PC}_w(G) - \text{PC}_{w+u}(G)|_2 \le \frac{\gamma}{4}$

*where*

$$\sigma = \frac{\gamma(1-\varepsilon)^l}{100M(l+2)\hat{\beta}^{l+1}\sqrt{2h\ln 4(lh+2)}}$$

*and* $M$ *is from Lemma 6.*

*Proof.* (Tropp, 2012) states that if $U \sim \mathcal{N}_{h^2}(0, \sigma^2 I)$, then

$$\Pr\left(||U||_2 \ge t\right) \le 2h\text{exp}\left(-\frac{t^2}{2h\sigma^2}\right)$$

and that if $V \sim \mathcal{N}_h(0, \sigma^2 I)$, then

$$\Pr\left(|V|_2 \ge t\right) \le 2\text{exp}\left(-\frac{t^2}{2h\sigma^2}\right)$$

So, applying it to our case:

$$\Pr\left(||U_1||_2 \le t \,\&\, ... \,\&\, ||U_l||_2 \le t \,\&\, |U^\varphi|_2 \le t \,\&\, |U^\omega|_2 \le t\right) \ge$$

$$\ge 1 - \sum_{i=1}^{l} \Pr\left(||U_i||_2 \ge t\right) - \Pr\left(|U^\varphi|_2 \ge t\right) - \Pr\left(|U^\omega|_2 \ge t\right) \ge$$

$$\ge 1 - 2(hl+2)\text{exp}\left(-\frac{t^2}{2h\sigma^2}\right)$$

We want this probability to be at least $\frac{1}{2}$. This condition satisfies when $t = \sigma\sqrt{2h\ln 4(lh+2)}$.

If $t = \sigma\sqrt{2h\ln 4(lh+2)}$ and $||U_i||_2 \le \frac{1}{l}\beta$, then with probability $\frac{1}{2}$ we have (Lemma 6):

$$|\text{PC}_w(G) - \text{PC}_{w+u}(G)|_2 \le eM\beta^{l+1}\left(|U^\varphi|_2 + |U^\omega|_2 + \sum_{i=1}^{l}||U_i||_2\right) \le$$

$$\le eM\beta^{l+1}(l+2)t \le eM\beta^{l+1}(l+2)\sigma\sqrt{2h\ln 4(lh+2)} \le$$

$$\le \frac{\hat{\beta}^{l+1}}{(1-\varepsilon)^{l+1}}eM(l+2)\sigma\sqrt{2h\ln 4(lh+2)} \le \frac{\gamma}{4}$$

It is left to check that we can apply Lemma 6: $||U_i||_2 \le \frac{1}{l}\beta$.

$$||U_i||_2 \le t = \frac{(1-\varepsilon)^{l+1}\gamma}{100\hat{\beta}^{l+1}(l+2)M} \le \frac{(1-\varepsilon)^{l+1}\gamma}{100(1-\varepsilon)^{l+1}\beta^{l+1}(l+2)M} \le \frac{\gamma\beta}{100M(l+2)\beta^{l+2}} \le$$

$$\le \frac{\gamma\beta}{100M(l+2)\left(\frac{\gamma}{2M}\right)^{\frac{l+2}{l+2}}} = \frac{1}{l+2}\frac{\beta}{50} \le \frac{1}{l}\beta$$

$\square$

Our next result establishes the bound for $\beta$s that falls into the interval: $|\beta - \hat{\beta}| < \frac{1}{l}\beta$

**Lemma 10.** *Let* $w = vec\{W_1, ..., W_l, W^\varphi, W^\omega\}$. *Let* $\hat{\beta}, \gamma \in \mathbb{R}_+$ *and* $\beta = \max\{||W_1||_2, ..., ||W_l||_2, |W^\varphi|_2 + 1\}$. *If* $w$ *such that* $|\beta - \hat{\beta}| \leq \varepsilon\beta$ *and* $\beta \geq \left(\frac{\gamma}{2M}\right)^{\frac{1}{l+2}}$, *then for any* $0 < \delta < 1$, *with probability at least* $1 - \delta$ *over an i.i.d size-$m$ training set $S$ according to $D$:*

$$L_{D,0}(\text{PC}_w) \leq L_{S,\gamma}(\text{PC}_w) + \mathcal{O}\left(\sqrt{\frac{M^2 l^2 h \ln lh \left(\frac{1+\varepsilon}{1-\varepsilon}\right)^{2l} \beta^{(2l+1)}|w|_2^2 + \ln\frac{m}{\delta}}{\gamma^2 m}}\right) \quad (7)$$

*Proof.* Let us use Lemma 1 with $Q(w + u) = \mathcal{N}(w, \sigma^2 I)$ and $P = \mathcal{N}(0, \sigma^2 I)$ where

$$\sigma = \frac{\gamma(1 - \varepsilon)^l}{100 M (l+2) \hat{\beta}^{l+1} \sqrt{2h \ln 4(lh + 2)}}$$

Thanks to Lemma 6, all conditions for application of Lemma 1 are satisfied.

Then, using Lemma 7 we can derive the upper bound on KL-Divergence:

$$D_{KL} = \frac{|w|_2^2}{2\sigma^2} = |w|_2^2 \mathcal{O}\left(\frac{M^2 l^2 \hat{\beta}^{2(l+1)} h \ln(lh)}{(1 - \varepsilon)^{2l} \gamma^2}\right) =$$

$$= \mathcal{O}\left(\left(\frac{1 + \varepsilon}{1 - \varepsilon}\right)^{2l} \frac{M^2 l^2 \beta^{2(l+1)} |w|_2^2 h \ln lh}{\gamma^2}\right)$$

Plugging this into the overall expression, we get:

$$L_{D,0}(\text{PC}_w) \leq L_{S,\gamma}(\text{PC}_w) + \mathcal{O}\left(\sqrt{\frac{M^2 l^2 h \ln lh \left(\frac{1+\varepsilon}{1-\varepsilon}\right)^{2l} \beta^{(2l+1)}|w|_2^2 + \ln\frac{m}{\delta}}{\gamma^2 m}}\right)$$

$\square$

The following lemma provides a lower bound on *interesting* values of $\beta$.

**Lemma 11.** *Let* $w = vec\{W_1, ..., W_l, W^\varphi, W^\omega\}$. *Let* $\beta = \max\{||W_1||_2, ..., ||W_l||_2, |W^\varphi|_2 + 1\}$.

*If* $w$ *such that* $\beta \leq \left(\frac{\gamma}{2M}\right)^{\frac{1}{l+2}}$, *then for any* $j > 0$ *and any* $G \in \mathcal{G}$,

$$|\text{PC}_w(G)[j]| \leq \frac{\gamma}{2}$$

*where $M$ from Lemma 6.*

*Proof.*

$$|\text{PC}_w(x)[j]| \leq |\text{PC}_w(x)|_2 \leq \prod_{i=1}^{l} ||W_i||_2 \, M(|W^\varphi| + 1) \leq \beta^{l+1} M \leq \beta^{l+2} M \leq \frac{\gamma}{2}$$

$\square$

This lemma provides us with the upper bound on the *interesting* values of $\beta$.

**Lemma 12.** *If* $w$ *such that* $\beta \geq (\gamma\sqrt{m})^{\frac{1}{l+2}}\left(\frac{1}{M^{\frac{1}{l+2}}} + 1\right)$, *then*

$$\sqrt{\frac{l^2 M^2 h \ln lh |w|_2^2 \beta^{2(l+1)} \left(\frac{1+\varepsilon}{1-\varepsilon}\right)^{2l} + \ln\frac{lMm}{\delta}}{\gamma^2 m}} \geq 1$$

*Proof.* First, we handle the case when $\gamma\sqrt{m} < 1$:

$$\sqrt{\frac{l^2 M^2 h \ln lh |w|_2^2 \beta^{2(l+1)} \left(\frac{1+\varepsilon}{1-\varepsilon}\right)^{2l} + l \ln \frac{Mm}{\delta}}{\gamma^2 m}} \geq 1 \Leftarrow \frac{l^2 M^2 h \ln lh |w|_2^2 \beta^{2(l+1)} \left(\frac{1+\varepsilon}{1-\varepsilon}\right)^{2l} + l \ln \frac{Mm}{\delta}}{\gamma^2 m} \geq$$

$$\geq \frac{l \ln \frac{Mm}{\delta}}{\gamma^2 m} \geq 1$$

From now on, we consider $\gamma\sqrt{m} > 1$. Let us lower-bound $|w|_2^2$:

$$|w|_2^2 = |W^\varphi|_2^2 + |W^\omega|_2^2 + \sum_{i=1}^{l} ||W_i||_F^2 \geq \frac{1}{l+2}\left(|W^\varphi|_2 + |W^\omega|_2 + \sum_{i=1}^{l} ||W_i||_F\right)^2 =$$

$$= \frac{1}{l+2}\left(|W^\varphi|_2 + 1 - 1 + |W^\omega|_2 + \sum_{i=1}^{l} ||W_i||_F\right)^2 \geq \frac{1}{l+2}(\beta-1)^2 \geq \frac{(\beta-1)^2}{l}$$

Now, we can plug this into the original expression:

$$\sqrt{\frac{l^2 M^2 h \ln lh |w|_2^2 \beta^{2(l+1)} \left(\frac{1+\varepsilon}{1-\varepsilon}\right)^{2l} + l \ln \frac{Mm}{\delta}}{\gamma^2 m}} \geq 1 \Leftarrow \frac{l^2 M^2 h \ln lh |w|_2^2 \beta^{2(l+1)} \left(\frac{1+\varepsilon}{1-\varepsilon}\right)^{2l} + l \ln \frac{Mm}{\delta}}{\gamma^2 m} \geq$$

$$\geq \frac{l^2 M^2 \beta^{2(l+1)}(\beta-1)^2}{l\gamma^2 m} \geq M^2 \frac{(\beta-1)^{2(l+1)}}{\gamma^2 m} \geq$$

$$\geq M^2 \frac{(\beta - (\gamma\sqrt{m})^{\frac{1}{2(l+1)}})^{(l+2)}}{\gamma^2 m} = 1$$

$\square$

**Lemma 4.** *Let $W^\varphi$ be the parameters of the point transformation in* PERSLAY*. Then, we have that*

$$\forall G \in \mathcal{G} \quad |\text{PERSLAY}_w(\mathcal{D}(G))|_2 \leq B_1 |W^\varphi|_2 + C_1 \leq M_1(|W^\varphi|_2 + 1)$$

*where*

$$A_1 = \begin{cases} \max_{G \in \mathcal{G}} card(\mathcal{D}(G)) & \text{if } \text{AGG} = \text{sum} \\ 1 & \text{if } \text{AGG} = \text{mean } or \text{ } k\text{-max} \end{cases}, \quad (B_1, C_1) = \begin{cases} (\sqrt{2}, \sqrt{8q}\,b) & \text{if } \varphi = \Lambda \\ (0, \sqrt{q}) & \text{if } \varphi = \Gamma \\ (\sqrt{3}(b+1), 0) & \text{if } \varphi = \Psi \end{cases}$$

*and $M_1 = A_1 \max\{B_1, C_1\}$.*

*Proof.*

$$|\text{PERSLAY}_w(\mathcal{D}(G))|_2 = |\text{AGG}\{\!\!\{\omega_w(p)\varphi_w(p) \mid p \in \mathcal{D}(G)\}\!\!\}|_2 \leq$$

$$\leq A_1 \left|\max_{p \in \mathcal{D}(G)} \omega_w(p)\varphi_w(p)\right| \leq A_1 \left|\max_{p \in \mathcal{D}(G)} \varphi_w(p)\right|$$

We denote by $A_1$ the following number: if AGG is sum, then $A_1$ is the maximum cardinality of the persistent diagram, otherwise (if AGG is $k$-max, mean) we set $A_1$ equals to 1. The last inequality comes from the fact that $|\omega| \leq 1$

Now we handle multiple choices of $\varphi$. Note that we denote $a[i]$ as $i$th element of the vector $a$.

- $\varphi = \Lambda$.

$$\max_{p \in \mathcal{D}(G)} \varphi_w(p)[i] = \max_{p \in \mathcal{D}(G)} \max\{0, p[2] - |t_i - p[1]|\} \leq 2b + |t_i|$$

$$\Rightarrow$$

$$|\text{PERSLAY}_w(G)|_2 \leq A_1 \left[\sum_{i=1}^{q} (2N + |t_i|)^2\right]^{1/2} \leq A_1 \left[\sum_{i=1}^{q} 8b^2 + 2|t_i|^2\right]^{1/2} \leq A_1 \left(\sqrt{8q}b + \sqrt{2}|W^\varphi|_2\right)$$

- $\varphi = \Gamma$.

$$\max_{p \in \mathcal{D}(G)} \varphi_w(p)[i] = \max_{p \in \mathcal{D}(G)} \exp\left(-\frac{|p[1] - t_i[1]|^2 + |p[2] - t_i[2]|^2}{2\tau^2}\right) \leq 1$$
$$\Rightarrow$$
$$|\text{PERSLAY}_w(\mathcal{D}(G))|_2 \leq A_1 \left[\sum_{i=1}^{q} 1\right]^{1/2} \leq A_1 \sqrt{q}$$

- $\varphi = \Psi$.

$$\max_{p \in \mathcal{D}(G)} \varphi_w(p)[i] = \max_{p \in \mathcal{D}(G)} p[1]t_i[1] + p[2]t_i[2] + t_i[3] \leq (b+1)(|t_i[1]| + |t_i[2]| + |t_i[3]|)$$
$$\Rightarrow$$
$$|\text{PERSLAY}_w(\mathcal{D}(G))| \leq A_1 \left[\sum_{i=1}^{q} (b+1)^2 (|t_i[1]| + |t_i[2]| + |t_i[3]|)^2\right]^{1/2} \leq$$
$$\leq A_1(b+1) \left[\sum_{i=1}^{q} 3(|t_i[1]|^2 + |t_i[2]|^2 + |t_i[3]|^2)\right]^{1/2} \leq \sqrt{3} A_1 (b+1) |W^\varphi|_2$$

$\square$

**Lemma 5.** *Let $w = vec\{W^\varphi, W^\omega\}$ and $u = vec\{U^\varphi, U^\omega\}$, where $U^\varphi$ and $U^\omega$ denote small perturbations on the parameters of the point transformations and weight functions of PERSLAY, respectively. Then:*

$$|\text{PERSLAY}_w(\mathcal{D}(G)) - \text{PERSLAY}_{w+u}(\mathcal{D}(G))|_2 \leq M_2(|W^\varphi|_2 + 1)(|U^\varphi|_2 + |U^\omega|_2)$$

*where*

$$A_2 = \begin{cases} \max_{G \in \mathcal{G}} card(\mathcal{D}(G)) & \text{if AGG = sum} \\ 3 & \text{if AGG = mean or } k\text{-max} \end{cases}, \quad B_2 = \begin{cases} 1 & \text{if } \varphi = \Lambda \\ \frac{1}{\tau e^{1/2}} & \text{if } \varphi = \Gamma, \\ \sqrt{3}(b+1) & \text{if } \varphi = \Psi \end{cases}$$

$M_2 = A_2 \max\{B_2, M_1 Lip(\omega)\}$, *and $Lip(\omega)$ is a Lipschitz constant of $\omega$.*

*Proof.*

$$|\text{PERSLAY}_w(\mathcal{D}(G)) - \text{PERSLAY}_{w+u}(\mathcal{D}(G))|_2 =$$
$$= |\text{AGG}\{\!\{\varphi_w(p)\omega_w(p) \mid p \in \mathcal{D}(g)\}\!\} - \text{AGG}\{\!\{\varphi_{w+u}(p)\omega_{w+u}(p) \mid p \in \mathcal{D}(G)\}\!\}|_2$$

Thanks to Lemma 8 and the fact that sum is less than the number of elements multiplied by a max of elements, we can derive the following upper bound:

$$|\text{PERSLAY}_w(\mathcal{D}(G)) - \text{PERSLAY}_{w+u}(\mathcal{D}(G))|_2 \leq$$
$$\leq A_2 \max_{p \in \mathcal{D}(G)} |\omega_w(p)\varphi_w(p) - \omega_{w+u}(p)\varphi_{w+u}(p)|_2$$

We set $A_2$ to the maximum number cardinality of the persistence diagrams if AGG is sum, or to 3 if AGG is $k$-max, mean

$$\max_{p \in \mathcal{D}(G)} |\omega_w(p)\varphi_w(p) - \omega_{w+u}(p)\varphi_{w+u}(p)|_2 \leq$$
$$\leq \max_{p \in \mathcal{D}(G)} \{|\omega_{w+u}(p)||\varphi_{w+u}(p) - \varphi_w(p)|_2 + |\varphi_w(p)|_2|\omega_{w+u}(p) - \omega_w(p)|\} \leq$$
$$\leq \max_{p \in \mathcal{D}(G)} |\varphi_{w+u}(p) - \varphi_w(p)|_2 + \max_{p \in \mathcal{D}(G)} |\varphi_w(p)|_2|\omega_{w+u}(p) - \omega_w(p)|$$

All of the considered *point transformation* functions are Lipschitz in terms of parameters. Let us make the same assumptions for the *weight* part. Then (note that we denote $a[i]$ as $i$th element of the vector $a$)

- $\varphi = \Lambda$.

$$\text{Lip}(\varphi) = 1 \Rightarrow \max_{p \in \mathcal{D}(G)} |\varphi_{w+u}(p) - \varphi_w(p)|_2 \leq \left[\sum_{i=1}^{q} |U^\varphi[i]|_2^2\right]^{1/2} = |U^\varphi|$$

- $\varphi = \Gamma$.

$$\text{Lip}(\varphi) = \frac{1}{\tau e^{1/2}} \Rightarrow \max_{p \in \mathcal{D}(G)} |\varphi_{w+u}(p) - \varphi_w(p)|_2 \leq$$

$$\leq \left[\sum_{i=1}^{q} \left(\frac{1}{\tau e^{1/2}}\right)^2 (U^\varphi[2i]^2 + U^\varphi[2i+1]^2)\right]^{1/2} = \frac{1}{\tau e^{1/2}} |U^\varphi|_2$$

- $\varphi = \Psi$.

$$\text{Lip}(\varphi) \leq (b+1) \Rightarrow \max_{p \in \text{Dg}(G)} |\varphi_{w+u}(p) - \varphi_w(p)|_2 \leq$$

$$\leq \left[\sum_{i=1}^{q} (b+1)^2 (U^\varphi[3i] + U^\varphi[3i+1] + U^\varphi[3i+2])^2\right]^{1/2} \leq \sqrt{3}(b+1)|U^\varphi|_2$$

Let us introduce the auxiliary variable $E$, which equals the coefficient in front of $|U^\varphi|$.

To sum it up:

$$|\text{PERSLAY}_w(\mathcal{D}(G)) - \text{PERSLAY}_{w+u}(\mathcal{D}(G))|_2 \leq A_2(B_2|U^\varphi|_2 + M_1\text{Lip}(\omega)(|W^\varphi|_2 + 1)|U^\omega|_2) \leq$$
$$\leq M_2(|W^\varphi|_2 + 1)(|U^\varphi|_2 + |U^\omega|_2)$$

where $M_2 = A_2 \max\{B_2, M_1\text{Lip}(\omega)\}$

$\square$

**Lemma 6.** *Let $w = vec\{W_1, ..., W_l, W^\varphi, W^\omega\}$ and $u = vec\{U_1, ..., U_l, U^\varphi, U^\omega\}$, where $U_i$ is the perturbation of $i$th linear layer of* PC, $U^\varphi$ *is the perturbation of the* point transformation *part, and $U^\omega$ is the perturbation of the* weight *part of* PERSLAY. *Also, let $T \geq \max\{||W_1||_2, ..., ||W_l||_2, |W^\varphi|_2 + 1\}$ and $\forall i : ||U_i||_2 \leq \frac{1}{l}T$, then we can derive the following upper bound:*

$$|\text{PC}_w(x) - \text{PC}_{w+u}(x)|_2 \leq eM T^{l+1} \left(|U^\varphi|_2 + |U^\omega|_2 + \sum_{i=1}^{l} ||U_i||_2\right)$$

*where $M = \max\{M_1, M_2\}$ from Lemmas 4 and 5 and e is the Euler's constant.*

*Proof.* Let us prove that:

$$|\text{PC}_w(G) - \text{PC}_{w+u}(G)|_2 \leq \left(1 + \frac{1}{l}\right)^l M T^{l+1} \left(|U^\varphi|_2 + |U^\omega|_2 + \sum_{i=1}^{l} ||U_i||_2\right)$$

and then using the fact that $\left(1 + \frac{1}{l}\right)^l \leq e$ we derive the desired bound. Note that $\psi_i$ is the activation function after $i$th layer and we use the fact that $\text{Lip}(\psi_i) \leq 1$ and $\psi_i(x) \leq |x|$ (Assumption 3).

$$
\begin{aligned}
|\mathrm{PC}_w^l(G) - \mathrm{PC}_{w+u}^l(G)|_2 &= \left| W_l\psi_l\left(\mathrm{PC}_w^{l-1}(G)\right) - (W_l + U_l)\psi_l(\mathrm{PC}_{w+u}^{l-1}(G)) \right|_2 = \\
&= \left| (W_l + U_l)\left(\psi_l(\mathrm{PC}_w^{l-1}(G)) - \psi_l(\mathrm{PC}_{w+u}^{l-1}(G))\right) - U_1\psi_l(\mathrm{PC}_w^{l-1}(G)) \right|_2 \leq \\
&\leq ||W_l + U_l||_2 \, |\mathrm{PC}_w^{l-1}(G) - \mathrm{PC}_u^{l-1}(G)|_2 + ||U_l||_2 \, |\mathrm{PC}_w^{l-1}(G)|_2 \leq \\
&\leq \left(1 + \frac{1}{l}\right) T \, |\mathrm{PC}_w^{l-1}(G) - \mathrm{PC}_{w+u}^{l-1}(G)|_2 + ||U_l||_2 \, |\mathrm{PC}_w^{l-1}(G)|_2 \leq \\
&\leq \left(1 + \frac{1}{l}\right) T \, |\mathrm{PC}_w^{l-1}(G) - \mathrm{PC}_{w+u}^{l-1}(G)|_2 + ||U_l||_2 \prod_{i=1}^{l-1}||W_i||_2 \, |\mathrm{PERSLAY}_w(G)|_2 \leq \\
&\leq \left(1 + \frac{1}{l}\right) T \, |\mathrm{PC}_w^{l-1}(G) - \mathrm{PC}_{w+u}^{l-1}(G)|_2 + ||U_l||S^{l-1}|\mathrm{PERSLAY}_w(G)|_2
\end{aligned}
$$

We want to prove this theorem using induction on the number of layers. So, we must prove the upper bound for the $l = 1$ case. Note that $\mathrm{PC}_w^0 = \mathrm{PERSLAY}_w$.

Using Lemmas 4 and 5 we can derive the upper bound for $l = 1$ case:

$$
\begin{aligned}
|\mathrm{PC}_w^1(G) - \mathrm{PC}_{w+u}^1(G)|_2 &\leq \left(1 + \frac{1}{l}\right) TM_2(|W^\varphi|_2 + 1)(|U^\varphi|_2 + |U^\omega|_2) + ||U_1||_2 M_1(|W^\varphi|_2 + 1) \leq \\
&\leq \left(1 + \frac{1}{l}\right) T^2 \max\{M_1, M_2\}(|U^\varphi|_2 + |U^\omega|_2 + ||U_1||_2)
\end{aligned}
$$

Now we want to prove the transition from $l$ to $l + 1$:

$$
|\mathrm{PC}_w^{l+1}(G) - \mathrm{PC}_{w+u}^{l+1}(G)|_2 \leq \left(1 + \frac{1}{l+1}\right)^{l+1} M \, T^{l+2}\left(|U^\varphi|_2 + |U^\omega|_2 + \sum_{i=1}^{l}||U_i||_2\right)
$$

$$
+||U_{l+1}||_2 T^l |\mathrm{PERSLAY}_w(G)|_2
$$

$$
\leq \left(1 + \frac{1}{l+1}\right)^{l+1} MT^{l+2}\left(|U^\varphi|_2 + |U^\omega|_2 + \sum_{i=1}^{l+1}||U_i||_2\right)
$$

$$\square$$

**Theorem 1.** *Let $w = vec\{W_1, ..., W_l, W^\varphi, W^\omega\}$ and $M = \max\{M_1, M_2\}$ from Lemmas 4 and 5. Then for any $\gamma, \delta > 0$ with probability at least $1 - \delta$ over i.i.d size-m training set S according to D, for any $W_1, ..., W_l, W^\varphi, W^\omega$, we have:*

$$
L_{D,0}(\mathrm{PC}_w) \leq L_{S,\gamma}(\mathrm{PC}_w) + \mathcal{O}\left(\sqrt{\frac{l^2 M^2 h \ln(lh)|w|_2^2 \beta^{2(l+1)} + \ln\frac{lMm}{\delta}}{\gamma^2 m}}\right)
$$

*where $\beta = \max\{||W_1||_2, ..., ||W_l||_2, |W^\varphi|_2 + 1\}$*

*Proof.* For $w$ such that $\beta \leq \left(\frac{\gamma}{2M}\right)^{\frac{1}{l+2}}$ the stated inequality holds, since in this case $L_{S,\gamma}(\mathrm{PC}_w) = 1$ due to Lemma 11. For $w$ such that $\beta \geq (\gamma\sqrt{m})^{\frac{1}{l+2}}\left(\frac{1}{M^{\frac{1}{l+2}}} + 1\right)$ the inequality also holds, since LHS always $\leq 1$ and RHS is $\geq 1$ in this case due to Lemma 12.

So, the interval of $\beta$s to consider is the following:

$$
\left(\frac{\gamma}{2M}\right)^{\frac{1}{l+2}} \leq \beta \leq (\gamma\sqrt{m})^{\frac{1}{l+2}}\left(\frac{1}{M^{\frac{1}{l+2}}} + 1\right)
$$

Let us choose $r = \varepsilon \left(\frac{\gamma}{2M}\right)^{\frac{1}{l+2}}$ to be a radius of the covering $C$ of this interval. Let us upper bound the size of this covering.

$$|C| \leq (\gamma\sqrt{m})^{\frac{1}{l+2}} \left(\frac{1}{M^{\frac{1}{l+2}}} + 1\right) \frac{1}{r} = (\gamma\sqrt{m})^{\frac{1}{l+2}} \left(\frac{1}{M^{\frac{1}{l+2}}} + 1\right) \frac{(2M)^{\frac{1}{l+2}}}{\varepsilon\gamma^{\frac{1}{l+2}}} = \mathcal{O}(l(M\sqrt{m})^{\frac{1}{l+2}})$$

we set $\varepsilon = \frac{1}{lM^{\frac{1}{l+2}}}$.

Therefore, denoting the event of Eq. 7 with $\hat{\beta}$ taking the $i$-th value of the covering and $\delta = \frac{\delta}{|C|}$ as $E_i$, we have:

$$\mathbb{P}\left(E_1 \& ... \& E_{|C|}\right) \geq 1 - \sum_{i=1}^{|C|} \mathbb{P}(\overline{E_i}) \geq 1 - |C|\frac{\delta}{|C|} = 1 - \delta$$

So, with the probability $1 - \delta$, we have that

$$L_{D,0}(\text{PC}_w) \leq L_{S,\gamma}(\text{PC}_w) + \mathcal{O}\left(\sqrt{\frac{l^2 M^2 h \ln lh \left(\frac{1+\varepsilon}{1-\varepsilon}\right)^{2l} |w|_2^2 \beta^{2(l+1)} + \ln \frac{m|C|}{\delta}}{\gamma^2 m}}\right) =$$

$$= L_{S,\gamma}(\text{PC}_w) + \mathcal{O}\left(\sqrt{\frac{l^2 M^2 h \ln lh |w|_2^2 \beta^{2(l+1)} + \ln \frac{lMm}{\delta}}{\gamma^2 m}}\right)$$

Note that $\left(\frac{1+\varepsilon}{1-\varepsilon}\right)^{2l} = \mathcal{O}(1)$ with $\varepsilon = \frac{1}{lM^{\frac{1}{l+2}}}$ $\qquad\qquad\square$

## C IMPLEMENTATION DETAILS

### C.1 DATASETS

Table 4 reports summary statistics of the datasets used in this paper.

Table 4: Statistics of the datasets.

| Dataset | #graphs | #classes | Avg #nodes | Avg #edges |
|---|---|---|---|---|
| NCI1 | 4110 | 2 | 29.87 | 32.30 |
| IMDB-B | 1000 | 2 | 19.77 | 96.53 |
| PROTEINS (full) | 1113 | 2 | 39.06 | 72.82 |
| MUTAG | 188 | 2 | 17.93 | 19.79 |
| DHRF | 756 | 2 | 42.43 | 44.54 |

### C.2 MODELS

We run all experiments using PyTorch. We closely follow the filtration functions used in (Carrière et al., 2020). In particular, we use Kernel heat functions with parameter $t = 0.1$ for MUTAG and PROTEINS and $t = 10$ for the remaining datasets. Instead of processing each diagram type using separate models, we combine ordinary and extended diagrams for 0- and 1-dimensional features and apply a single model. After obtaining persistence diagrams, we discard node features. We use mean aggregation function in all experiments, and Gaussian point transformations. For the feedfoward part of PersLay, we apply ReLU activation functions. All models are trained with Adam (Kingma & Ba, 2015) and learning rate of $10^{-3}$ for 3000 epochs.

**Dependence on model paramaters.** Regarding the dependence on the spectral norm of weights, we reported results for a model with a final MLP (multilayer perceptron) of 2 hidden layers (3 layers in total) and width of 128 for all layers. The number of parameters of the point transformation was $q = 100$. For the experiments on width vs. generalization gap, we used $q = 100$, and 1 hidden layer with a varying width in $\{32, 64, 128, 256, 512\}$.

**Point transformations.** Here, we used an MLP with 2 hidden layers with 128 neurons each. The number of parameters of the point transformation was $q = 400$. Moreover, we applied a max aggregation function.

**Regularizing PersLay.** For the experiments regarding ERM and spectral norm regularizers, we perform model selection for $l \in \{2, 3\}$, and $\lambda \in \{10^{-3}, 10^{-4}, 10^{-5}, 10^{-6}\}$. Again, we use Gaussian point transformation, $q = 100$, and width equals to 128. Our goal was to see if we could observe gains from the regularized version even for shallow neural networks.

**Hardware.** For all experiments, we use Tesla V100 GPU cards and consider a memory budget of 32GB of RAM.

## D   ADDITIONAL VISUALIZATIONS

Figure 8 and Figure 9 report additional results for the triangle point transformation on the three largest datasets: PROTEINS, NCI1, and IMDB-BINARY. In particular, Figure 8 shows the dependence of the generalization on width, while Figure 9 shows the dependence on the spectral norm. Overall, our bound can capture the trend in the empirical gap and produces high correlation values for all datasets.

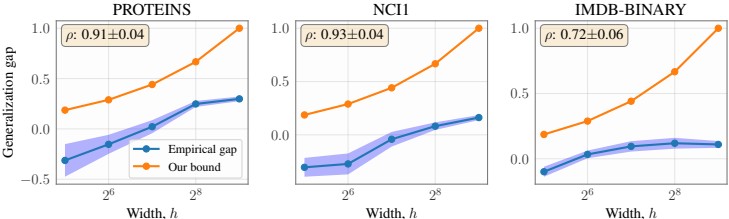

Figure 8: **Width vs. generalization gap for the triangle point transformation.**

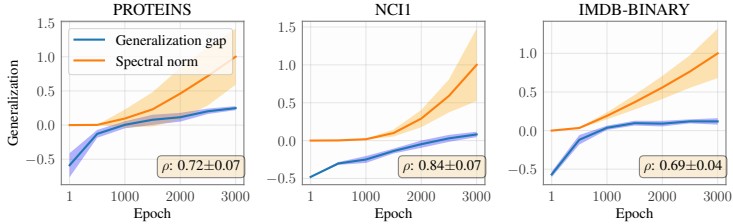

Figure 9: **Spectral norm vs. generalization gap for the triangle point transformation.**