# OpenReview forum: "How well does Persistent Homology generalize on graphs?"
_ICLR.cc/2024/Conference — Submitted to ICLR 2024_

### Official Review · Reviewer_yvKz · 2023-10-28

**Soundness:** 3 good
**Presentation:** 2 fair
**Contribution:** 2 fair
**Rating:** 3
**Confidence:** 3

**Summary:**

The authors analyze the generalization (prediction beyond training set) power of persistent homology on graphs. They also generalize existing vectorization techniques by adding non-linear layers and give experimental studies on graph classification tasks.

**Strengths:**

The authors study an important problem how good the persistent homology is in prediction tasks in graph representation learning. They explore several approaches to answer this question and give experimental analysis.

**Weaknesses:**

1. While the question addressed in this paper is important, its contribution appears to be relatively modest, especially when compared to the recent work by Morris et al. (2023). In that study, the authors delve into the predictive power of Graph Neural Networks (GNNs) through the lens of VC-dimension, and the authors of the current paper adapt their methods to the context of Persistent Homology. While the theoretical results are intriguing, their practical relevance in the machine learning domain remains questionable.

The provided bounds are often complex and abstract, making them challenging to compute in real-world applications. Given the extensive, nine-page proof section, which requires a thorough review, the paper might be better suited for a journal focused on statistics or applied topology rather than an ML venue. The heavy theoretical content, with limited applicability, calls into question the practical utility of the paper's findings in the ML community.

2. The paper's readability and coherence could be significantly improved, as it currently suffers from the need for clearer definitions and explanations of key concepts. The exposition and the paper's overall objective should be more explicitly stated to facilitate a better understanding of its content.

**Questions:**

Figures 3 and 4 are very interesting. In Figure 4, while the width is increasing, empirical values of the generalization gap stay very low. How do we explain this? On the other side, could you provide insights into the relationship between the generalization gap and the training set size?

---

> ### Author Response · Authors · 2023-11-17
>
> Thanks for your review. Below, we address your questions/comments.
>
> > "contribution appears to be relatively modest, especially when compared to the recent work by Morris et al. (2023) [...] the authors of the current paper adapt their methods to the context of Persistent Homology."
>
> **Our focus and analysis are orthogonal to Morris et al. (2023).** Thanks for the chance to clarify the relationship between our Proposition 2 and the result by Morris et al. (in Proposition 1). We highlight that we only use the work by Morris to show an analogous result --- we don't build upon it or use it to prove anything. Also, our work reports a PAC-Bayesian analysis, which is unrelated to Morris et al. We added a plot to depict the dependence between the Lemmas used to prove our Proposition 2. We hope this clarifies the orthogonality of our work wrt to Morris'.
>
> > "While the theoretical results are intriguing, their practical relevance in the machine learning domain remains questionable...The provided bounds are often complex and abstract, making them challenging to compute in real-world applications."
>
> **Our theoretical results are important in their own right.** Understanding the generalization behavior of ML algorithms, i.e., analyzing their ability to predict well once trained,  is one of the main goals of machine learning in general and statistical machine learning in particular. From that perspective, our theoretical results are already significant.
>
> **Practical implications of our work.** We introduce a new algorithm by regularizing PersLay (please see the new Section 3.4). The regularization term directly exploits the dependence of parameters established in our bounds. In our experiments with several real-world experiments, we show that this method generally improves performance over the standard (i.e., unregularized) PersLay algorithm across many real datasets, often significantly. Our empirical results strongly substantiate the practical benefits of our theoretical analysis that leads to a new algorithm with better performance.
>
> > "Given the extensive, nine-page proof section, which requires a thorough review, the paper might be better suited for a journal focused on statistics or applied topology rather than an ML venue. The heavy theoretical content, with limited applicability, calls into question the practical utility of the paper's findings in the ML community."
>
> **"Theoretical issues in deep learning" is an explicitly identified relevant topic for ICLR (https://iclr.cc).** Although there has been a surge in PH-based methods for machine learning, our understanding of their theoretical properties is vastly under-explored, especially for graph-structured data. Therefore, we believe that our work represents an important contribution to the topological deep learning community.
>
> > "The paper's readability and coherence could be significantly improved [...] and the paper's overall objective should be more explicitly stated to facilitate a better understanding of its content."
>
> **We have revised the presentation to make this work broadly accessible**. Thanks for your comment. We have rewritten sections of the paper for clarity. In particular, we have
> - clarified the relevance of the VC-Dimension bounds in the context of generalization and expressivity of any PH method (Section 3.1);
> - fixed typos and included a notation table in the Appendix;
> - expanded the notation, including the abbreviation (kFWL for k Folklore WL);
> - clarified which results apply to PH in general vs those that apply to PersLay (now called PersLay Classifier);
> -added a related works section (Section 1.2)
> - added the dependence between the different Lemmas for the VC-Dim bound (Figure 2);
> - clarified how our PAC-Bayes generalization bound for PersLay differs from the corresponding bounds for graph neural networks and feedforward networks;
> - refined the presentation by re-writing parts of sections 2 and 3 for clarity and precision.
>
> > "Figures 3 and 4 are very interesting. In Figure 4, while the width is increasing, empirical values of the generalization gap stay very low. How do we explain this?"
>
> First, we note that the empirical gap is a function of many quantities, including those other than width. More importantly, we have observed strong Pearson correlation coefficients between the empirical and theoretical gap across datasets, validating our analysis.
>
> > "could you provide insights into the relationship between the generalization gap and the training set size?"
>
> The generalization gap decreases with, i.e., is inversely proportional to, the square root of the training set size. Theorem 2 reports this relationship where $m$ is the training set size.
>
> ---
> Thanks for your review. We improved the readability and coherence of our paper based on your feedback. If your concerns are sufficiently solved, we would appreciate if you consider raising your score. Otherwise, we will be happy to engage further and provide more clarifications.

---

> > ### Comment · Reviewer_yvKz · 2023-11-18
> >
> > Thank you for the revision. While I appreciate the theoretical perspective in your results, I'm having trouble seeing their practical application in machine learning based on your experiments. I'm still concerned that the experimental section is relatively weaker, whereas the nine-page proof section constitutes the primary contribution of the paper. Therefore, I believe a thorough mathematical review for accuracy in the proofs is necessary. Due to these reasons, I'm inclined to keep my score.

---

> ### Author Response · Authors · 2023-11-19
> **Thank you. Theoretical underpinnings of PH for ML have been lagging behind.**
>
> Many thanks for acknowledging our response, and sharing your perspective.
>
> It is our considered opinion that theoretical underpinnings provide a strong foundation that enables future empirical success stories. Good generalisation in particular has been one of the most important cornerstones and quests of Machine Learning from its very inception. In fact, generalization is at the core of the classical Structural Risk Minimization (SRM) principle that subsumes some of the most successful ML algorithms including SVMs (and kernel methods), Lasso, Ridge Regression, etc. SRM hinges on finding a good regularizer to guard against overfitting by controlling model complexity.
>
> Despite a surge in applying PH methods for ML applications, the theoretical foundations of these methods are grossly underexplored [1, 2].  We presented here the first generalisation bounds for PH and introduced a novel structural risk minimisation algorithm (regularized PH method) based on our bounds that led to significantly improved performance on several real datasets, paving way for more comprehensive empirical investigations in domains such as drug discovery where PH descriptors are increasingly being employed [3].
>
> Besides, our work has laid concrete foundations for analysing generalisation ability of other PH methods, and determining appropriate regularizers for principled design of novel practical algorithms for graph representation learning. We believe this shall provide a much-needed fillip to the topological data analysis and deep learning community.
>
> Again, thank you for your feedback. We'd love to discuss any further thoughts or comments you might have.
>
> We do hope for, and would greatly appreciate, your stronger support for this work once our theoretical proofs are vetted by some subset of reviewers during the ongoing review process.
>
> [1] Turkes et al. On the Effectiveness of Persistent Homology, NeurIPS (2022).
>
> [2] Immonen et al. Going beyond persistent homology using persistent homology, NeurIPS (2023).
>
> [3] Demir et al. ToDD: Topological Compound Fingerprinting in Computer-Aided Drug Discovery, NeurIPS (2022).

---

### Official Review · Reviewer_ys8K · 2023-10-31

**Soundness:** 3 good
**Presentation:** 2 fair
**Contribution:** 3 good
**Rating:** 6
**Confidence:** 3

**Summary:**

This paper provides first theoretical bounds for generalization of persistent homology (PH) on graphs. The results are supported with an experimental study on 5 real-world graph classification benchmarks. Moreover, additional experiment illustrates how the bounds can be used to regularize the PH pipeline.

**Strengths:**

Strengths are already described in the summary.

**Weaknesses:**

I did not identify any major weaknesses in the paper, although I also did not check all the proofs in the supplementary material. However, the clarity of presentation could be significantly improved, see my questions and comments below.

**Questions:**

(Q1) Related work: The related literature is missing. Does “generalization capabilities of PH methods remain largely uncharted territory” imply that there are no earlier works in this direction? If so, you can make it more explicit.  How is the recent paper [1] related to your work?

[1] Immonomen, Souza and Garg, Going beyond persistent homology using persistent homology

(Q2) Section 3.1: It is not very clear how Section 3.1 fits into the whole story of your paper, in particular since you write that expressivity and generalization can be at odds with each other, but you do not elaborate further. This issue is pronounced more in (the very nice) Figure 2, which is missing Proposition 1 and 2, and Lemma 2 and 3. What is the reason that Morris et al., 2023 and Rieck, 2023 that you rely on in Section 3.1, are not discussed in the Related work? Why do the experiments not validate these theoretical results too?

(Q3) Table 1: In the Discussion, you write the following: “In Table 1, the study provides a valuable resource by depicting the resulting bound dependencies on various parameters. This information is instrumental in estimating the overhead introduced by PH in the generalization performance of conventional multi-layer perceptron.” Immediately I was hoping to see some discussion in this direction, but I am not sure if the next two paragraphs are related to Table 1? Where does the $\sqrt{\ln b}$ appear, and where do we see $h \sqrt{\ln h}$? References to particular lemmas, theorems or tables can improve readability.

In general, can you provide some intuition about what makes the generalization bounds for PH different from other models, and/or what properties of PH do you use to obtain your theoretical results? Is it crucial that the input is a graph?

Moreover, could you summarize the “key insights about the limits and power of PH methods”? What do we learn from your paper about the generalization ability of PH?

(Q4) “We report additional results across different epochs and hyper-parameters in the supplementary material.” These results are not included?

(Q5) Figure 5 is not very informative. Could it be replaced by including the results for the line and triangle point transformations (and their correlations) into Figure 3 and Figure 4?

(Q6) Notation: The notation could be improved, what is the current logic? For example, you could e.g. use small case/capital case/Greek alphabet for graph nodes/sets/functions, and then be consistent. Often, you use the same notation for different things: e.g., S for both training set and the upper bound in Lemma 6, m for the size of training data and the maximal number of distinguishable graphs, omega for the PersLay weight function and for the hypothesis parameters, etc. Could you include a notation table? For instance, it took me quite some time to find what b is when seeing it appear at the end of Section 3. As you will see, a lot of the minor comments below would likely be resolved with a table summary of improved notation.


Minor comments:

-	In the paragraph on PAC-Bayesian Analysis, you define L_S, gamma and L_D, gamma before S, D, gamma and L are introduced. Also, for better clarity, the order of the formulas here should be reversed? Moreover, you use the notation L here, later within the line point transformation, and later also for a layer.
-	In the paragraph on the Analysis Setup, when writing h_1=q, reminder the reader briefly what q is, or at least reference Figure 1.
-	The acronym FWL is never introduced?
-	For the node with Lemma 1 in Figure 2, you could reference Neyshabur (2018) to make it clearer that this is an earlier result and not the contribution of this paper.
-	It is not clear to me when you use |x|, and when ||x||? Is ||.||_F from Table 1 ever described?
-	At the end of statement of Lemma 4, I suggest to reference “(see PersLay in Section 2)”, so that the reader can easily find what AGG and Phi, Lambda, Gamma and L are.
-	Add full stop at the end of Lemma 4 and Lemma 5.
-	Which norm do you use for persistence diagrams, i.e., what is |D(G)|?
-	What is e in Lemma 6?
-	Why do we see L_D, 0 in Theorem 2, what scenario does gamma=0 reflect, can you provide some intuition? Also, you start this theorem with “Let w  = …”, and then claim that “for any w, we have…”? On a related note, later in Section 4 you write that generalization gap is measured as L_D,0 – L_S, gamma=1, but you do not provide more info?
-	Does the description of h, d and W_i in the caption of Table 1 reflect only the third row, or the complete table? In the latter case, make this description a separate sentence. Do you also want to mention again also what q, l, e, beta are?
-	We compute correlation coefficients between -> We compute [specify which] correlation coefficients \rho between
-	“alpha is a hyperparameter that balances the influence of the two terms”: alpha should probably be replaced with lambda?
-	“Our research highlights the significance of leveraging the principles of regularization to enhance the performance of machine learning models across diverse applications.” I found this sentence rather surprising (not the focus of this work), could you rephrase or elaborate?

---

> ### Author Response · Authors · 2023-11-17
> **Response: Part 1/3**
>
> Thanks for your detailed and thoughtful review. Below, we address your questions/comments.
>
> > "Related work: The related literature is missing. Does “generalization capabilities of PH methods remain largely uncharted territory” imply that there are no earlier works in this direction? If so, you can make it more explicit. How is the recent paper [1] related to your work?"
>
> Thank you for your thoughtful feedback, and the opportunity to emphasize the contributions of this work.
>
> **First generalization results for PH-based methods.** Indeed, to the best of our knowledge, the generalization bounds established here are first such results for PH. Previously, PH has been used to analyze generalization behavior of other ML methods; however, the generalization of PH itself has not been considered in literature prior to this work.
>
> **Positioning wrt Immonen et al (NeurIPS'23).** Thank you for pointing us to the important work by Immonen et al (NeurIPS 2023). Our contributions are different from theirs.
>
> From a theoretical perspective, they focus on the {\em expressivity} of PH methods,  characterizing the exact class of graphs that can be recognized by PH methods that rely on color-based filtrations.
> They do not consider generalization of PH methods at all, which in contrast is the focus of this work.
>
> From a practical perspective, they propose a new, more expressive method RePHINE that yields improved performance on several real-world datasets. We leverage the dependence of parameters in our generalization bounds to propose a new method that regularizes PersLay, also achieving improvement on real data.
>
> Indeed, both expressivity and generalization are important from an ML perspective, and designing algorithms that strike a right balance between expressivity and generalization is a central goal of statistical learning theory. Thus, our work should be viewed as complementary to Immonen et al.
>
> > "Section 3.1: It is not very clear how Section 3.1 fits into the whole story of your paper, in particular since you write that expressivity and generalization can be at odds with each other, but you do not elaborate further."
>
> **Expressivity vs. generalization, and the relevance of Section 3.1** Thank you for the opportunity to clarify this.
>
> Indeed, enhancing expressivity typically comes at the expense of  generalization. Morris et al. (2023) established such a result for Graph Neural Networks, showing that the  VC-dimension of GNNs with $L$ layers is lower bounded by the maximal number of graphs that can be distinguished by 1-WL (Weisfeiler-Leman test for isomorphism).  High VC-dimension directly translates to poor generalization, whereas by definition   greater the number of graphs that can be distinguished greater the expressivity. Thus, Morris et al. showed the tension between expressivity and generalization in the context of message passing GNNs that are at most as expressive as the 1-WL test.
>
> However, these results do not apply to PH. In section 3.1, we fill this gap by establishing the conflict between expressivity and generalization for any PH method, building on the results about expressivity of PH by another influential paper due to Rieck (2023). Importantly, we connect the generalization of PH via VC-dimension to expressivity in terms of the WL hierarchy (i.e., 1-WL and more expressive higher order WL tests). Our results in Section 3.1 hold particular significance since (topological descriptors obtained from) PH methods are increasingly being used to augment the capabilities of message-passing GNNs, so being able to analyze one in terms of the other is a step forward.
>
> The result in Section 3.1 provides a lower bound on the generalization for any PH method, and thus complements the lower bound on expressivity by Rieck (2023) - both in terms of the WL/Folklore WL (FWL) hierarchy. Albeit important, these bounds do not take into account the underlying data distributions. Therefore, in the rest of the paper, we develop a data-dependent PAC-bayes bound on generalization of PersLay, a flexible and widely used PH method.
>
> Importantly, unlike the lower VC-dim bounds in Section 3.1 that are hard to estimate, we provide an upper bound on the generalization error in terms of more amenable parameters that directly lead to a new regularized PersLay method with demonstrable empirical benefits on real-world datasets.
>
> > "This issue is pronounced more in (the very nice) Figure 2, which is missing Proposition 1 and 2, and Lemma 2 and 3."
>
> Thanks for catching this. We have added a subplot to Figure 2, depicting the dependence between the Lemmas (2 and 3) used to prove Proposition 2.
>
> > "What is the reason that Morris et al., 2023 and Rieck, 2023 that you rely on in Section 3.1, are not discussed in the Related work?"
>
> Thanks for pointing this out. We have added a related works sections (Section 1.2), where we also discuss these works.

---

> ### Author Response · Authors · 2023-11-17
> **Response: Part 2/3**
>
> > "you write the following: “In Table 1, the study provides [...] bound dependencies on various parameters. This information is instrumental in estimating the overhead introduced by PH in the generalization performance of conventional multi-layer perceptron.” Immediately I was hoping to see some discussion in this direction, but I am not sure if the next two paragraphs are related to Table 1?"
>
> **Overhead due to PersLay.** Thanks for your bringing our attention to this. Based on your comment, we have improved the discussion regarding the discrepancy between our bound for the PersLay Classifier and the bounds for feedforward networks and GNNs. The dependence on width and spectral norms of the weights directly stems from Theorem 2 by absorbing non-related  quantities within the big-O notation. Analyzing the width dependence we conclude that the upper bound for PersLay increases from $O(\sqrt{h \ln h})$ (in feedforward networks and GNNs) to $O(h\sqrt{\ln h})$ when $q = \Theta(h)$. So, to produce a tighter generalization bound,
> we recommend choosing $q = o(h)$. We have also clarified this in the revised version.
>
> >  "Where does the lnb appear, and where do we see hlnh? References to particular lemmas, theorems or tables can improve readability."
>
> **Regarding $\ln b$ and $h \ln h$.** Here, $b$ denotes the radius of the $\ell_2$-ball in which the points of the persistence diagrams lie. The dependence on $b$ is subsumed within the quantity $M$ that appears in Theorem 2 (please, see Lemmas 4 and 5).  The additional dependence (compared to feedforward networks and GNNs) of our bound on the maximum input norm is proportional to $\sqrt{\ln b}$, which is upper bounded by $b$. Thus, PH does not incur any overhead due to this term, and so we have removed this now from the discussion section for improved exposition.
>
> For $h \ln h$, again, this appears in Theorem 2. Please note that, unlike the dependence on $b$, PersLay incurs an overhead proportional to $O(\sqrt{h})$ when $q = \Theta(h)$, as clarified above.
>
> > "In general, can you provide some intuition about what makes the generalization bounds for PH different from other models, and/or what properties of PH do you use to obtain your theoretical results? Is it crucial that the input is a graph?"
>
> The main sources of difference come from the choice of the persistence diagram vectorization (Lemmas 4 and 5) and from the combination of perturbations in vectorization and linear layers (Lemma 6).
>
> Regarding the applicability to graphs, we note that our results in Section 3.1 regarding the VC dimension applies to graph data while  the PAC-Bayes bound applies more generally to diagrams extracted from any input.
>
> > "Moreover, could you summarize the “key insights about the limits and power of PH methods”? What do we learn from your paper about the generalization ability of PH?"
>
> Some **key insights from our analysis** are:
> - There is an inherent tension between generalization and expressivity of any PH method in the context of graph representation learning (both can be lower bounded in terms of the WL hierarchy). In particular, enhancing expressivity leads to an increase in the VC-dimension, thereby worsening the ability to generalize (Section 3.1).
> - A flexible PH method, namely PersLay, has provably good generalization performance. In particular, the dependence of generalization on margin, sample complexity (i.e., size of the training dataset), and maximum norm across points in the persistence diagrams resembles and is comparable to that for feedforward neural networks (FNNs, Neyshabur et al., 2018) and graph neural networks (GNNs, Liao et al., 2020).
> - A key difference between generalization of PersLay and FNNs/GNNs is the dependence on the model width and the spectral norm of the model weights. Specifically, PersLay has slightly worse dependence in terms of these quantities; however, this gap can be regulated by modulating the dimensionality of the embedding of the persistence diagram before treating with the linear layers.
> - From a practical viewpoint, the theoretical dependence of generalization on parameters can be leveraged to introduce a regularized method in a structural risk minimization setup that leads to improved empirical performance on several real-world datasets.
>
> > "“We report additional results across different epochs and hyper-parameters in the supplementary material.” These results are not included?"
>
> Apologies for this. We did not find significant variability in these experiments and ended up not including them.
>
> > "Figure 5 is not very informative. Could it be replaced by including the results for the line and triangle point transformations (and their correlations) into Figure 3 and Figure 4?"
>
> Thanks for the suggestion. We will remove Figure 5 and replace it with additional plots (like Fig. 3 and 4) for line and triangle transformations. We will update the paper with this change by the end of this discussion period.

---

> ### Author Response · Authors · 2023-11-17
> **Response: Part 3/3**
>
> > "Notation: The notation could be improved, what is the current logic? For example, you could e.g. use small case/capital case/Greek alphabet for graph nodes/sets/functions, and then be consistent. Often, you use the same notation for different things: e.g., S for both training set and the upper bound in Lemma 6, m for the size of training data and the maximal number of distinguishable graphs, omega for the PersLay weight function and for the hypothesis parameters, etc. Could you include a notation table? For instance, it took me quite some time to find what b is when seeing it appear at the end of Section 3. As you will see, a lot of the minor comments below would likely be resolved with a table summary of improved notation."
>
> **We have revised the presentation**. We have rewritten sections of the paper for clarity. In particular, we have
> - clarified the relevance of the VC-Dimension bounds in the context of generalization and expressivity of any PH method (Section 3.1);
> - fixed typos and included a notation table in the Appendix;
> - expanded the notation, including the abbreviation (kFWL for k Folklore WL);
> - clarified which results apply to PH in general vs those that apply to PersLay (now called PersLay Classifier);
> - added a related works section (Section 1.2)
> - added the dependence between the different Lemmas for the VC-Dim bound (Figure 2);
> - clarified how our PAC-Bayes generalization bound for PersLay differs from the corresponding bounds for graph neural networks and feedforward networks;
> - refined the presentation by re-writing parts of sections 2 and 3 for clarity and precision.
>
> **Importantly, we have accepted all your minor suggestions/comments and implemented the changes in the revised manuscript.** We really appreciate your careful reading.
>
> ---
>
> Many thanks for your thoughtful and incisive comments. We hope our response has elucidated several subtle points that you raised and allayed your concerns. We would greatly appreciate your stronger support for this work.

---

> > ### Comment · Reviewer_ys8K · 2023-11-19
> >
> > I have read the responses to all of the reviewers, and I very much appreciate the effort that the authors made to address each of the concerns. I skimmed through the revised documents, but I haven't read them in detail; if most of the comments added here in the discussion are incorporated in the paper (it seems so), I think the work is definitely improved and I support its acceptance to ICLR.
> >
> > Some final minor comments: Consider revising the notation table once again, e.g. h_i (as a row in it's own right), beta, gamma, b, q ... are missing? I asked you what |D(G)| is, i.e. which norm do you use for persistence diagrams, but I think this information has not yet been included in the main text? I had to look it up in the proof of Lemma 4, where you write that, in case AGG=sum, "A1 is the maximum size of the persistent diagram", but what is size?

---

> > > ### Comment · Reviewer_ys8K · 2023-11-19
> > >
> > > I would also like to add that I find this comment you made extremely useful:
> > >
> > > "The main sources of difference come from the choice of the persistence diagram vectorization (Lemmas 4 and 5) and from the combination of perturbations in vectorization and linear layers (Lemma 6)."
> > >
> > > but I don't think you include this discussion in the revised paper? In my first round of feedback, I asked a lot about some intuitive summary about what we learn from your paper about the generalization of PH, and I think a paragraph about the influence of b, and the choice of point transformation, AGG and q would be very valuable.

---

> ### Author Response · Authors · 2023-11-19
>
> We are grateful for your active engagement and all the additional feedback. We summarize below how we've included all of your suggestions and comments.
>
> > Regarding the notation |D(G)| ---size of the diagram.
>
> We have updated the paper to clarify this notation. Instead of using $|\mathcal{D}(G)|$ to denote the number of points in the persistence diagrams, we are now using card($\mathcal{D}(G)$) and call it the cardinality (instead of size) of the diagram $\mathcal{D}(G)$. We added the following sentence to the first paragraph of page 3:
>
> *We denote the persistence diagram for a graph $G$ as $\mathcal{D}(G)$ and use $\text{card}(\mathcal{D}(G))$ to represent its cardinality, i.e., number of (birth time, death time) pairs in $\mathcal{D}(G)$.*
>
>  We have also updated the notation table accordingly.
>
> > Consider revising the notation table once again, e.g. h_i (as a row in it's own right), beta, gamma, b, q ... are missing?
>
> Thanks for catching this. We have included the following entries in the notation table:
> - $q$: dimensionality of the PersLay embeddings
> - $h_i$: dimensionality of the input to the $i$-th MLP layer
> - $\gamma$: margin scalar in the margin-based loss
> - $b$: the maximum norm of the input to the feedforward network of the PersLay classifier
> - $\beta$, $\hat{\beta}$: $\max\\{||W_1||_2, ... ||W_l||_2, |W^{\varphi}|_2 + 1\\}$, arbitrary approximation of $\beta$
>
> We've also improved some descriptions and the presentation of the Notation table.
>
> > I would also like to add that I find this comment you made extremely useful: "The main sources of difference [...] combination of perturbations in vectorization and linear layers (Lemma 6)." but I don't think you include this discussion in the revised paper?
>
> We have now included this discussion in Section 3.4 (paragraph 'Comparison to other bounds').
>
> > I asked a lot about some intuitive summary about what we learn from your paper about the generalization of PH, and I think a paragraph about the influence of b, and the choice of point transformation, AGG and q would be very valuable.
>
> Thanks to your comment, we have extended our discussion section by adding a paragraph about the 'Influence of PersLay' in Section 3.3 (Discussion). The new paragraph reads like this:
>
> **Influence of PersLay components.** *Our analysis shows that when $AGG= \text{sum}$, it is hard to obtain reasonable generalization guarantees since $M$ depends on the cardinality of the persistence diagram, which can be large. Also, we can conclude that the generalization performance of PC using the line point transformation has a weaker dependence on $q$ ($M_1, M_2$ does not depend on $q$ in this case) than other point transformations. This is particularly relevant if one chooses a large value for $q$.*
>
> In our previous answer, we have listed some key take-aways from our work. Here, we list them again alongside with pointers to where you can find the corresponding discussion in the paper:
> - There is an inherent tension between generalization and expressivity of any PH method in the context of graph representation learning (both can be lower bounded in terms of the WL hierarchy). In particular, enhancing expressivity leads to an increase in the VC-dimension, thereby worsening the ability to generalize (Section 3.1).  --- **We have added this right before Proposition 2**;
> - A flexible PH method, namely PersLay, has provably good generalization performance (analyzed here in the context of classification). In particular, the dependence of generalization on margin, sample complexity (i.e., size of the training dataset), and maximum norm across points in the persistence diagrams resembles and is comparable to that for feedforward neural networks (FNNs, Neyshabur et al., 2018) and graph neural networks (GNNs, Liao et al., 2020). --- **this discussion appears in the paragraph  'Comparison to other bounds' (Section 3.3)**;
> - A key difference between generalization of PersLay and FNNs/GNNs for classification is the dependence on the model width and the spectral norm of the model weights. Specifically, PersLay Classifier has slightly worse dependence in terms of these quantities; however, this gap can be regulated by modulating the dimensionality of the embedding of the persistence diagram before treating with the linear layers. --- **this discussion also appears in the paragraph 'Comparison to other bounds' (Section 3.3). We also added sentence about the main sources of differences.**
> - From a practical viewpoint, the theoretical dependence of generalization on parameters can be leveraged to introduce a regularized method in a structural risk minimization setup that leads to improved empirical performance on several real-world datasets. --- **we have added a separate subsection to introduce the regularized PersLay (Section 3.4).**
>
> Many thanks!

---

> > ### Author Response · Authors · 2023-11-22
> >
> > Dear Reviewer ys8k,
> >
> > We are indebted for your thorough and invaluable feedback, as well as insights and constructive comments, throughout the review process that have helped us significantly in elucidating several subtle aspects of this work and position its contributions appropriately.
> >
> > Much gratitude also for actively engaging in discussions with other reviewers towards building a consensus.  We are grateful for your incisive inputs and continued support throughout the discussion phase!
> >
> > Could you kindly acknowledge that all your concerns and comments have been addressed with the latest version of the paper, or if there is anything else we could discuss or clarify to improve your evaluation of the work?  Thank you so much.
> >
> > Best regards!

---

> > > ### Author Response · Authors · 2023-11-23
> > >
> > > Dear reviewer ys8k,
> > >
> > > As a final comment, we would like to note we have added new visualizations regarding the dependence of our bound on the width and the spectral norm for the **triangle point transformation** (Appendix D). In the final version of our paper, we will replace Fig. 5 with these additional plots along with ones for the line transformation, as you suggested.
> > >
> > > Best regards.

---

### Official Review · Reviewer_qwyi · 2023-11-01

**Soundness:** 3 good
**Presentation:** 1 poor
**Contribution:** 2 fair
**Rating:** 6
**Confidence:** 2

**Summary:**

In this paper, the generalization performance of persistent homology is given in terms of PAC-Bayes. Normalized margine bounds are given via PersLay, a method that encompasses various vectorizations of the Persistent diagram. Normalized margine bounds has been theoretically proven and experiments have confirmed the theorem.

**Strengths:**

- Theoretical derivation and proof of normalized margine bounds for persistent diagrams of graphs are given.
- It is basically an analogy to Neyshabur et al. (2018), but combined with PersLay, which encompasses vectorization of various persistent diagrams to apply to persistent homology.

**Weaknesses:**

The purpose of the main theorem seems unclear. Although the theory claims a theory of generalizability of the persistent homology of graphs, what the actual theorem shows is a generalization bound for maps that combine PersLay, ReLu, and DNN in the persistent homology. In fact, Neyshabur et al. (2018) explicitly states that it gives generalization bounds for DNNs. There seems to be a gap between the generalization performance of Persistent homology and the generalization bounds of PH, the mapping. Also, PH is only an example of one network and not for a general network.Currently, it appears to be a derivation of the generalization boundary of a self-defined network. Whether one is arguing for generalization bounds for persistent homology itself or for generalization bounds for networks using persistent homology, it seems to me that the arguments need to be organized and additional discussion is needed.

The biggest complaint is that it is extremely reader-unfriendly. For example, the definition of $gamma$-margine loss was written some time after its first appearance. It doesn't even say what k-FWL is; it may be Folklore Weisfeiler-Lehman, but it is not self-evidently recognizable to all readers of the subject. Map PH seems to be the entire architecture of Fig. 1, but the definition is unclear and the caption of Fig. 1 is sometimes described as PersLay's architecture, making it difficult to grasp.

**Questions:**

Please comment on the above.

---

> ### Author Response · Authors · 2023-11-17
>
> Thanks for your feedback. In the following, we address all your questions/comments.
>
> > "Although the theory claims a theory of generalizability of the persistent homology of graphs, what the actual theorem shows is a generalization bound for maps that combine PersLay, ReLu, and DNN in the persistent homology."
>
> **Our VC-dimension results hold for any PH method on graphs.** We note that Proposition 2 sets a lower bound on the VC-dim of *any* PH method that distinguishes graphs based on their persistence diagrams obtained from arbitrary filtration functions. In particular, this establishes that enhancing expressivity of PH methods comes at the expense of generalization. Importantly, this result also connects the generalization of PH via VC-dimension to expressivity in terms of the WL hierarchy (i.e., 1-WL and more expressive higher order WL tests), which is the primary tool for analyzing and designing new GNN architectures.
>
> **Relevance of PersLay: it accommodates flexible vectorizations broadly used for PH.** Although important for showcasing the tension between generalization and expressivity, the VC-Dimension bounds do not take into account the underlying data distributions. Therefore, we establish the first data-dependent PAC-Bayesian generalization bounds fora flexible and widely used PH-based framework, namely Perslay. In particular, PersLay subsumes most persistence diagram vectorizations (e.g., persistence landscapes, persistence silhouette, deep sets, persistence images, and Gaussian-based vectorizations) proposed in the literature. Finally, our work also lays a strong foundation for analyzing other classes of PH methods, including those adopting learnable filtrations.
>
> > "PH is only an example of one network and not for a general network.Currently, it appears to be a derivation of the generalization boundary of a self-defined network. [...] it seems to me that the arguments need to be organized and additional discussion is needed."
>
> **The network design is flexible.** Thanks for giving us the opportunity to clarify and improve the presentation. We analyzed multiple persistence diagram vectorization approaches in combination with a very general classification head (feedforward networks), following the same setup as PersLay. In fact, our choice of using PersLay for generalization analysis was motivated by the generality of the PersLay's vectorization scheme, which subsumes most of the previously proposed ones. These schemes have been also used in more recent methods, such as TOGL [Topological GNN, ICLR'22]. Moreover, PersLay is one of the most influential PH-based models for graph data, which is the focus of this work. Based on your feedback, we have clarified this in the revised version of the manuscript.
>
> > "The biggest complaint is that it is extremely reader-unfriendly. For example, the definition of gamma-margine loss was written some time after its first appearance. It doesn't even say what k-FWL is; it may be Folklore Weisfeiler-Lehman"
>
> **We have revised presentation to make this work broadly accessible.** Thanks for your comment. We have rewritten sections of the paper for clarity. In particular, we have
>  - clarified the relevance of the VC-Dimension bounds in the context of generalization and expressivity of any PH method (Section 3.1);
>  - fixed typos and included a notation table in the Appendix;
>  - expanded the notation, including the abbreviation (kFWL for k Folklore WL);
>  - clarified which results apply to PH in general vs those that apply to PersLay (now called PersLay Classifier);
>  - added a related works section (Section 1.2)
>  - added the dependence between the different Lemmas for the VC-Dim bound (Figure 2);
>  - clarified how our PAC-Bayes generalization bound for PersLay differs from the corresponding bounds for graph neural networks and feedforward networks;
>  - refined the presentation by re-writting parts section 2 and 3 for clarity and precision.
>
> We thank you for your inputs, and believe the revised version is accessible for a broader audience and fully addresses the issues you pointed out.
>
> > "Map PH seems to be the entire architecture of Fig. 1, but the definition is unclear and the caption of Fig. 1 is sometimes described as PersLay's architecture, making it difficult to grasp."
>
> Apologies for the confusion. We have modified the caption to clarify that it depicts PersLay followed by multilayer perceptrons, now called PersLay Classifier (PC).
>
> ---
> Thank you for your constructive feedback. The quality of presentation has significantly improved based on your comments. We hope that our answers have sufficiently addressed your concerns, and will appreciate if you consider increasing your score. If you need any further clarification, please let us know.

---

> > ### Comment · Reviewer_qwyi · 2023-11-19
> >
> > Thank you for your revision and clarifications.
> > I am aware from the outset that PersLay encompasses a lot of filtration and vectorization, and my concern was that ReLu and DNN were fixed. I understood this point to be arguing that generalization is only generalization as a classifier, and by showing this with respect to ReLu+DNN, the most classical method, it is possible to show that vectorization using TDA retains generalization ability, i.e., if you choose a good classifier, you can show generalization ability. I would like to raise my score by one as the writing is much improved. However, since the theorem itself is limited to fixed classifiers, the scope of this contribution is still narrow and its impact on many users of ML with TDA has not yet been felt. I feel further research on this point is needed.

---

> > > ### Comment · Reviewer_ys8K · 2023-11-19
> > >
> > > "Although the theory claims a theory of generalizability of the persistent homology of graphs, what the actual theorem shows is a generalization bound for maps that combine PersLay, ReLu, and DNN in the persistent homology." and the concern about the ReLU and DNN being fixed are very important, and I agree. Now that the authors differentiate between PH, PersLay and PersLay Classifier, I think the exposition is clearer and made more precise. Taking the relevance of PersLay into account, I think the first theoretical result about the generalization of a PH-based approach is nice (first step in the right direction).

---

> > > > ### Comment · Reviewer_qwyi · 2023-11-19
> > > >
> > > > Thank you for your comment. Yes, I agree that this theoretical result is first step in the right direction. In this regard, I have a positive impression. I think it is a question of what level of quality is required as an ICLR accepted paper. This is not something that is clear-cut, though.Since this is a relative discussion, I would like to take into account the opinions of other reviewers, including AC.

---

> > > > > ### Author Response · Authors · 2023-11-20
> > > > >
> > > > > Dear reviewers qwyi and ys8k,
> > > > >
> > > > > Thank you so much for sharing further insights. We appreciate your time and excellent inputs.
> > > > >
> > > > > Reviewer qwyi:
> > > > >
> > > > > Thanks for your positive response -  we are glad that the improved presentation (writing) helped position our work better.
> > > > >
> > > > > Our choice for analyzing the architectural design of the PersLay classifier was fully driven by i) the proposed model by Carriere et al. (2020) and ii) the ubiquitous choice of MLPs as the classification component of deep models trained end-to-end (e.g., GNNs, CNNs, etc). While the former ensures our analysis holds for a broadly adopted model for persistence diagram vectorization, the latter abides by a common practice in DL. Naturally, since we are interested in generalization bounds, our analysis applies to supervised classification settings.
> > > > >
> > > > > We deliberated further over your concern about combining PersLay with a seemingly restricted downstream architecture. We revisited our analysis based on your feedback, and it has helped us generalize the scope to significantly more flexible architectures.
> > > > >
> > > > > Specifically, we noticed that our results do not require ReLU as the activation function and in fact, hold for several most commonly used activation functions. In particular, the only place where the role of activation shows up in our analysis is Lemma 6, where we use two properties of the activation function $\psi$, namely, (a) $\psi$ has Lipshitz constant at most 1, and (b) $\psi(x) \leq |x|$.  Thus our bounds hold not only for ReLU but also subsume **any combination of** most commonly used non-linear activations  such as ReLU, Leaky ReLU, SoftPlus, Tanh, Sigmoid, and max-pooling in the MLP. In fact, because of the Lipschitz property, it suffices to have $\psi(0) = 0$ to satisfy $\psi(x) \leq |x|$.
> > > > >
> > > > > Thanks to your feedback, we have now modified the paper to reflect this aspect of generality of our analysis. In particular, in the revised version of the paper, we have:
> > > > >  1. updated Fig. (1),
> > > > >  2. modified the description of the PersLay classifier, and
> > > > >  3.  added the following sentence to the assumptions (Section 3.2):
> > > > >
> > > > > *$\forall 1 \leq i \leq l \quad Lip(\psi_{i}) \leq 1, \psi_i(x) \leq |x|$. Thus our analysis subsumes any combination of most commonly used non-linear activation functions (e.g. ReLU, Leaky ReLU, SoftPlus, Tanh, Sigmoid, and max-pooling) in the MLP.*
> > > > >
> > > > > We're grateful for your constructive feedback that has led to significantly expanding the scope of our contribution. Thank you, and we hope this translates into your improved assessment of this work.

---

> > > > > > ### Comment · Reviewer_qwyi · 2023-11-21
> > > > > >
> > > > > > I think the discussion on ReLu further enhances the value of the proposed methodology. Since the discussion was not available at the time of submission, I leave it to the AC to decide whether to add this point to the evaluation. Generalization performance depends on the classifier, but I believe that if a simple linear classifier can be shown, it can be regarded as generalization performance at the feature level.

---

> > > > > > > ### Author Response · Authors · 2023-11-21
> > > > > > >
> > > > > > > Thank you for acknowledging the enhanced merits of this work. We would like to note that our analysis already subsumes a linear classifier:
> > > > > > >
> > > > > > > > Generalization performance depends on the classifier, but I believe that if a simple linear classifier can be shown, it can be regarded as generalization performance at the feature level.
> > > > > > >
> > > > > > > In our notation, $l$ corresponds to the number of layers of the classifier. Thus, to attain the generalization bound for PersLay + linear classifier, we set $l=1$ and choose the identity as the activation function. In that case, our bound (Theorem 1) reduces to
> > > > > > >
> > > > > > > $
> > > > > > > \mathcal{O}\left( \sqrt{\frac{M^2 h \ln(h) |w|_2^2 \beta^4  + \ln(Mm/\delta)}{\gamma^2 m}}\right)
> > > > > > > $
> > > > > > >
> > > > > > > where $w=\text{vec}(\\{W_1, W^\varphi, W^\omega\\})$, $W_1$ is the weight matrix of the linear classifier, and $\beta=\max\\{\||W_1\||_2, |W^\varphi|_2+1\\}$.
> > > > > > >
> > > > > > > ---
> > > > > > >
> > > > > > > Thank you for the opportunity to underscore this. We hope that this addresses all your concerns, and would greatly appreciate the same being reflected in an increased score.

---

> > > > > > > > ### Comment · Reviewer_qwyi · 2023-11-22
> > > > > > > >
> > > > > > > > This paper needs to revise its claim points and revise its writing, but I would like to raise my score in hopes of improvement.

---

> > > > > > > > > ### Author Response · Authors · 2023-11-22
> > > > > > > > >
> > > > > > > > > Dear Reviewer qwyi,
> > > > > > > > >
> > > > > > > > > Thank you for your detailed review and valuable feedback that has helped us improve the overall clarity of our presentation. We have carefully addressed your concerns, including expanding the scope of our analysis to subsume a wide range of commonly used activation functions and flexible architectures. We believe these enhancements reinforce the significance and merits of this work, and greatly appreciate your help during the review process.
> > > > > > > > >
> > > > > > > > > Best regards!

---

### Official Review · Reviewer_1CoA · 2023-11-01

**Soundness:** 3 good
**Presentation:** 3 good
**Contribution:** 3 good
**Rating:** 6
**Confidence:** 3

**Summary:**

This paper analyzes the generalization power of PersLay and derives new generalization bound. In addition, the paper discusses a VC-dim lower bound for persistent homology (PH) in terms of the WL-test on graphs. Experimental results demonstrate that the theoretical bounds can well capture the trend observed in the empirical generalization gap.

**Strengths:**

1. Perslay is an excellent model in TDA and it boosts the incorporation of PH with GNNs. Due to its effectiveness, Perslay has no theoretical guarantees. This paper provides new insights about the generalization of Perslay and provides new upper bound.
2. This paper extends the expressive power of PH in terms of WL to get a lower bound regarding the generalization ability of PH.

**Weaknesses:**

1. 1.The proofs are based on some assumptions, e.g. the filtrations of PH are fixed. However, many recent works [1][2] are based on flexible filtration function when using Perslay. Perslay itself is a powerful tool to vectorize persistence diagrams (PD) and provides informative representations. It can be plugged into many other models when using PH, and these models already have strong generalization power, such as GNNs. Therefore, analyzing the bound of Perslay or PH may not be necessary.
2. This paper merely investigated the generalization one vectorization tool of PD, i.e. Perslay, thus having limited contribution. Researchers who are interested in the generalization of PH on graphs may be more interested in other representations of PD, such as persistence images and deep sets [3], and in models with flexible filtrations [2].

[1] Hofer Christoph, et al. "Graph filtration learning." ICML 2020.

[2] Horn Max, et al. "Topological graph neural networks." ICLR 2022.

[3] Manzil Zaheer, et al. “Deep sets”, NIPS 2017.

**Questions:**

Please refer to the weakness.

---

> ### Author Response · Authors · 2023-11-17
>
> Thanks for your feedback. We reply to your comments/questions below.
>
> > "The proofs are based on some assumptions, e.g. the filtrations of PH are fixed. However, many recent works [1][2] are based on flexible filtration function when using Perslay."
>
> **Our VC-dimension result does not make any assumptions and holds for all PH methods.** Please note that our VC-dimension analysis (section 3.1) is not limited to fixed filtration functions. In particular, our analysis establishes that enhancing expressivity of PH methods comes at the expense of generalization. Importantly, this result also connects the generalization of PH via VC-dimension to expressivity in terms of the WL hierarchy, which is the primary tool for analyzing and designing new GNN architectures.
>
> **Fixed filtration functions dominate the PH/ML literature.** The widespread use of learnable function is a relatively recent phenomenon in PH-based ML, and usually runs orders of magnitude slower compared to non-learnable ones. Arguably, applying non-learnable functions still represents the mainstream approach in TDA.
>
> **Some works have explicitly advocated for fixed filtration functions (with learnable vectorizations) over learnable filtrations.** Filtration functions can come in different flavors; for instance, they can rely on node degree [1], cliques [2], or node attributes [3]. Some of the popular options are parameter-free. Also, while some works showed gains using learnable filtrations [4], others have reported no benefits and adopted fixed functions instead [5,6]. There is still no consensus about the significance of the gains associated with learnable filtration in many application.
>
> [1] Deep learning with topological signatures. NeurIPS 2017.
>
> [2] Networks and cycles: A persistent homology approach to complex networks. ECCS 2013.
>
> [3] Going beyond persistent homology using persistent homology. NeurIPS 2023.
>
> [4] Topological GNNs. ICLR 2022.
>
> [5] PersLay. AISTATS 2020.
>
> [6] Improving Self-supervised Molecular Representation Learning using Persistent Homology. NeurIPS 2023.
>
> > "It can be plugged into many other models when using PH, and these models already have strong generalization power, such as GNNs. Therefore, analyzing the bound of Perslay or PH may not be necessary."
>
> **Generalization guarantees of GNNs are not preserved when PH is incorporated.** While PH has been tightly integrated into graph base models (e.g., GNNs), this integration does not preserve the generalization capabilities of the base models. In particular, we know that the combination of GNNs and PH can distinguish a strictly larger class of graphs compared to GNNs alone. Therefore, we expect the VC-Dimension of GNN+PH to be greater than that of GNNs, and hence, significantly affecting the generalization guarantees of GNNs.
>
> **Understanding generalization of PH is essential to inform design choices (e.g., network architecture).** For instance, a key difference between generalization of PersLay and FNNs/GNNs is the dependence on the model width and the spectral norm of the model weights. Specifically, PersLay has slightly worse dependence in terms of these quantities; however, this gap can be regulated by modulating the dimensionality of the embedding of the persistence diagram before treating it with linear layers. Such important design choices are informed by our analyses.
>
> > "This paper merely investigated the generalization one vectorization tool of PD, i.e. Perslay, thus having limited contribution. Researchers [...] may be more interested in other representations of PD, such as persistence images and deep sets [3]"
>
> **Our analysis subsumes persistence images and deep sets.** We note that the vectorization methods introduced by PersLay are rather general and subsume the suggested persistence images (PI) and also DeepSet-based strategies. Thus, our analysis covers most of the vectorization schemes broadly used in the literature.
>
> **Our work lays a strong foundation for analyzing learnable filtrations.** One way to analyze PH with learnable filtration schemes could be to get upper bounds on perturbation of outputs in terms of the filtration function parameters. This would additionally require an analysis of Wasserstein distances between persistence diagrams obtained with different parameters. We believe that for a specific class of graphs we can get modified upper bounds for perturbation with respect to filtration function parameters that would depend on Wasserstein distance of same order. This additional analysis could be readily integrated into our framework to get generalization bounds for learnable filtrations.
>
> ---
>
> Thank you for your feedback. We believe that your comments have served to better position the importance of our contributions, including the significance of our results and the versatility of our analysis. We hope that our answers have sufficiently addressed your concerns, and would be grateful if they translate into an improved assessment of our work.

---

> > ### Comment · Reviewer_1CoA · 2023-11-22
> >
> > Thank you for your response. I have looked into other comments as well.
> > In general, my concerns are well addressed and I would like to raise my score.
> > Very interesting paper.

---

> > > ### Author Response · Authors · 2023-11-22
> > >
> > > Dear Reviewer 1CoA,
> > >
> > > Thank you so much for your insightful feedback. We are pleased to note that our responses have effectively addressed your concerns, and acknowledge your positive evaluation and recognition of this work as being very interesting. We greatly appreciate your time and attention in helping us underscore the contributions of this work.
> > >
> > > Best regards!

---

### Comment · Area_Chair_STSP · 2023-11-10
**Authors-Reviewers discussion starts today, ends on Nov 22**

Dear authors and reviewers,

@Authors: please make sure you make the most of this phase, as you have the opportunity to clarify any misunderstanding from reviewers on your work. Please write rebuttals to reviews where appropriate, and the earlier the better as the current phase ends on Nov 22, so you might want to leave a few days to reviewers to acknowledge your rebuttal. After this date, you will no longer be able to engage with reviewers. I will lead a discussion with reviewers to reach a consensus decision and make a recommendation for your submission.

@Reviewers: please make sure you read other reviews, and the authors' rebuttals when they write one. Please update your reviews where appropriate, and explain so to authors if you decide to change your score (positively or negatively). Please do your best to engage with authors during this critical phase of the reviewing process.

This phase ends on November 22nd.

Your AC

---

> ### Author Response · Authors · 2023-11-19
> **Thank you so much for this initiative!**
>
> Dear AC,
>
> We're grateful to you for your great service to our community, and for proactively orchestrating a meaningful discussion with the reviewers.
>
> We took some time to make sure we address all the comments, questions, and concerns raised by the reviewers, as well as incorporate their  (many constructive and excellent) suggestions to the best of our ability. We've turned in our responses to the individual reviews, and updated both the main and the supplementary files to reflect our commitment to improving the presentation of this work based on the feedback. We hope this facilitates our discussions going forward.
>
> Irrespective of how this process pans out,  we could not be more thankful for this kind gesture of yours.
>
> Best regards,
> Authors

---

> > ### Comment · Area_Chair_STSP · 2023-11-19
> >
> > Dear authors,
> >
> > Thank you for your kind note -- not sure if I deserve any special credit as this is the AC's job! Thanks for submitting rebuttals.
> >
> > AC

---

### Author Response · Authors · 2023-11-23
**Many thanks for your service**

We are grateful to all the reviewers for their time and insightful comments, as well as to the (senior) area, program, and general chairs for their service to the community.

We are glad that the reviewers note that our work is very interesting (1CoA), studies an important problem (yvKz), and is the first one to provide insights about the generalization of an excellent model in TDA (1CoA, ys8K). Also, our work presented theoretical derivations (qwyi) supported by an experimental study on 5 real-world graph classification benchmarks (ys8K).

To the best of our efforts, we've tried to address all the specific comments, including the minor ones, that have been raised by each reviewer. In particular, we have:

1. rewritten parts of the text for clarity, including a description of the PersLay-based classifier (Section 2);
2. clarified the relevance of the VC-Dimension bounds in the context of generalization and expressivity of any PH method (Section 3.1);
3. included a notation table in the Appendix (Appendix A);
4. clarified the versatility of our analysis, including, that it applies to persistence images and deep sets as well (Section 2);
5. added a related works section (Section 1.2);
6. added the dependence between the different Lemmas for the VC-Dim bound (Figure 2);
7. extended our analysis to flexible architectures subsuming any combination of several commonly used activation functions beyond ReLU (assumptions in Section 3.2);
8. elucidated the aspects that make PAC-Bayes bounds for PersLay in classification settings different and the analysis challenging compared to the corresponding bounds and analysis for both Feedforward Networks as well as Graph Neural Networks (Section 3.3);
9. improved the discussion section, highlighting the main takeaways from our work, including an effective design choice for good generalization (Section 3.3);
10. discussed the connections to structured risk minimization and emphasized the practical merits of our generalization bounds in informing a regularized model that performs significantly better on several real-world datasets (Section 3.4);
11. included additional visualizations regarding the dependence on width and spectral norm for the triangle point transformation (Appendix D).

We owe gratitude to the area chair and the reviewers for their active engagement during the review period, and believe that acting on the reviewers' excellent feedback has reinforced the many strengths of this work and improved its presentation. Thank you so much!

---

### Meta-Review · Area_Chair_STSP · 2023-12-05

**Metareview:**

This meta-review is a reflection of the reviews, rebuttals, discussions with reviewers and/or authors, and calibration with my senior area chair. This paper explores the generalisation performance of persistent homology through PAC-Bayes. There is a consensus among reviewers that despite its merits and the interesting ideas introduced, the paper is not easy to read and clarity could be significantly improved. Experiments are seen as somewhat sub-par. There is a sentiment that the paper is unlikely to have a large impact unless significantly revised. I appreciate this will come as a disappointment to the authors: ICLR is a highly competitive venue and unfortunately the clarity of this submission is sup-optimal.

**Justification For Why Not Higher Score:**

There is a consensus among reviewers that despite its merits and the interesting ideas introduced, the paper is not easy to read and clarity could be significantly improved. Experiments are seen as somewhat sub-par. There is a sentiment that the paper is unlikely to have a large impact unless significantly revised.

**Justification For Why Not Lower Score:**

N/A

---

### Decision · Program_Chairs · 2024-01-16

Reject